# Numerical and Experimental Investigation on Nosebleed Air Jet Control for Hypersonic Vehicle

Lin Zhang [1,*], Junli Yang [1], Tiecheng Duan [1], Jie Wang [1], Xiuyi Li [1] and Kunyuan Zhang [2]

1   Flight Technology College, Civil Aviation Flight University of China, Guanghan 618307, China; yjl_tom@163.com (J.Y.); duantc_cafuc@163.com (T.D.); wangjiecafuc@163.com (J.W.); lixiuyi6690@163.com (X.L.)
2   College of Energy and Power Engineering, Nanjing University of Aeronautics and Astronautics, Nanjing 210016, China; zkype@nuaa.edu.cn
*   Correspondence: harrisonzhang@163.com

**Abstract:** A new idea of nosebleed air jets with strong coupled internal and external flow is put forward using the lateral jet control principle to improve the maneuverability and fast reaction capabilities of hypersonic vehicles. The hypersonic vehicle's nose stagnant high-pressure and high-temperature gas is utilized as the drive source for long-term jet control. The significant coupled jet interaction of the internal and external flow changes the aerodynamic characteristics. As a result, the structure is basic and does not rely on any external source to achieve flight attitude control. The complicated flow characteristics of the nosebleed jet in supersonic crossflow surrounding the vehicle were numerically and experimentally investigated. The jet interaction characteristics and the aerodynamic characteristic changes generated by the nosebleed air jet are verified by comparing the flow field with and without the jet. Results indicate that the nosebleed air jet alters the center-of-pressure coefficient, which is subsequently coupled with the interference aerodynamic force. This results in a variation in pitch moment. The jet decreases the pitching moment coefficient when compared with the case without a jet. It is probable that combining nosebleed air jets with model centroid adjustment yields an optimal trim angle of attack.

**Keywords:** hypersonic vehicle; lateral jet; jet interactions; active flow control; wind tunnel test

## 1. Introduction

Lateral jet control, as an effective method of upper-level attitude control, uses direct force control formed by jet interactions to reduce the aerodynamic rudder surface and structural weight, and simplify the thermal protection design, which has become a trend in future aircraft development [1]. Compared with the traditional rudder control surface, the lateral jet control changes the flow field structure through the interaction between the jet and the mainstream, affects the aerodynamic characteristics, has high efficiency, fast response speed, and the reaction time is about 6~10 ms. A substantial amount of study has been carried out on jet interference/control [2–6]. The interference flow field is extremely sensitive to changes in incoming flow and jet flow due to the interaction of shock wave, expansion wave, and vortex. When properties such as jet location, pressure ratio, and velocity are combined with flight parameters such as Mach number and angle of attack, the result is a tremendously complicated external shock wave and vortex flow pattern. The bow shock surface caused by the local jet will extend to both sides of the nozzle position to generate interference aerodynamic force, which is superimposed with the force generated by the jet to form the actual lateral force. In particular, the jet interaction has an inherent instability trend as well as a possibility of abrupt changes in flow characteristics, and the flow characteristics are very nonlinear. To ensure the precision and stability of flight control, it is required to properly predict the change of the jet interaction flow field and aerodynamic characteristics, as well as increase the efficiency of jet control.

Researchers used the flat jet as the research object and studied the change of jet flow characteristics, the structure of jet interaction flow field and pressure distribution, the characteristics of the jet shear layer, the structure of large-scale vortex, the main factors affecting jet and the problem of multi-nozzle jet interaction through experiments, and verified the numerical simulations, such as the Reynolds averaged numerical simulation (RANS), large eddy simulation (LES), detached eddy simulation (DES), and hybrid RANS/LES methods [7–13]. Li [14] employed the Bayesian method to recalibrate the closure coefficient of the Spalart–Allmaras (SA) turbulence model in order to improve the performance of jet interaction in supersonic and quantify the uncertainty of separation length and wall pressure. However, because of the various variations between the actual situation and the flat jet interaction, using and popularizing the approximation model and prediction approach produced from the flat jet interaction is problematic.

The study on lateral jet control of many common practical challenges is carried out based on the understanding of the mechanics of flat jet interaction. DeSpirito [5] and Srivastava [15] used numerical simulations and experiments to investigate the influence of nozzle configuration, jet pressure ratio, and angle of attack on jet interaction. The turbulence model causes the vortex flow field to alter by more than 30% when there is an aerodynamic rudder surface. The jet interaction force varies by at least 6% depending on the angle of attack. The control effect is lessened when windward side jets form. Stahl [16] experimentally studied the flow characteristics and flow field structure changes under different pressure ratios and different gas jets. The jet pressure ratio has a significant impact on external flow field structure, and wall pressure distribution is affected by jet angle, jet pressure ratio, Reynolds number, etc. The parameters work together to excite the boundary layer and determine the size of the separation zone. Pudsey [11,17] used numerical simulations to investigate the jet interaction flow field and multi-jet interaction characteristics, as well as the influence of nozzle configuration and turbulence model on the flow field. Kitson [18] investigated the influence of jet-induced fluid–solid interaction on aircraft performance and stability. Through tests, Erdem [19] examined the sonic transverse jets at Mach 5 crossflow across a flat plate with a sharp leading edge. The interaction zone between the incoming crossflow developing on the flat plate and the jet was investigated, and jet penetration boundaries were defined. Zhang [20] and Zhen [21] employed the wedge–lateral jet interaction to improve the jet penetration height, lateral force, and jet control efficiency. Dong [22] employed numerical simulation of jet interaction and used wind tunnel data to verify its efficiency.

The impact of the jet on aerodynamic characteristics is highlighted. Graham [23] used the thin layer RANS and Euler techniques to investigate how a lateral jet interacts with the external flow for a variety of missile body geometries. The computational results were compared with results from a previously published wind tunnel study that primarily consisted of global force and moment measurements. The numerical techniques showed good agreement with the experiments at supersonic Mach numbers. Brandeis [24] investigated the lateral jet test of aircraft with various head shapes and discovered that the jet angle and nozzle shape influence response force. The effect of the jet on the lift surface was also investigated. The coupling effect causes the force or moment to dramatically vary, resulting in the additional aerodynamic force [25]. Grandihi [26] investigated the effect of the jet difference between a flat plate and a rotating body on pressure distribution and control efficiency. The factors influencing jet control efficiency were investigated, and it was discovered that increasing the jet pressure ratio improves control efficiency [27]. Kang [28] used CFD to model and evaluate the missile's aerodynamic database using a lateral jet and established two methods of simplified CFD considering all factors and the Latin hypercubic-Kriging surrogate model, which can better simulate the aerodynamic coefficients. The error is less than 10% when compared with that of the flight test results. Chai [29] examined the jet interaction characteristics between the waverider and the rotating body by applying the lateral jet to the waverider hypersonic vehicle. It was discovered that the waverider's interaction area is concentrated on the upper surface and difficult to expand to the side.



Xue [30] has proposed an innovative separation system that uses lateral jets to assist with store-safe separation, addressed at the problem of store separation from the internal cavity.

The current jet interaction problem primarily investigates the formation, development, and variation of the external flow field under a certain jet pressure ratio, and then analyzes the variation of aerodynamic characteristics. In particular, the jet pressure ratio is quite big, and the impact of outflow on the jet is often ignored. At the moment, lateral jet control technology mostly employs the high-temperature and high-pressure gas generated by the engine or carries extra high-pressure and high-density gas for the jets. It has the benefit of producing a direct lateral force that is unaffected by flight altitude, speed, or attitude. In practice, it will increase the complexity of the structure and extra weight. The center of gravity of the aircraft will be influenced by the consumption of engine fuel or a gas source. It typically works in pulse mode with a defined operating duration. When the fuel or gas supply runs out, it can no longer be used. When using solid fuel, the thrust cannot be adjusted. In this paper, a new idea of a nosebleed air jet with strong coupled internal and external flow is proposed for hypersonic vehicles with blunt heads in the atmosphere based on the lateral jet principle. This jet is powered by its own high-pressure gas and can continually jet on demand. Its structure is basic, and it does not require any additional power sources. The feasibility of employing a nosebleed air jet to control flight attitude is explored by investigating the variation of the nosebleed air jet induced by a change in flying attitude, the flow characteristics of strong coupled internal and external flow, and the formation and evolution of a complex flow field.

## 2. Model and Methodology

### 2.1. Concept of Nosebleed Air Jet

The nosebleed air jet idea (illustrated in Figure 1) utilizes the high-temperature and high-pressure airflow stagnated at the head of the hypersonic vehicle as the continuous jet's driving air source. The lateral jet is performed on the aircraft surface after the jet flow is blooded from the airflow stagnation zone. The separation shock wave generated by the jet deflects the shock wave caused by the aircraft head, resulting in the emergence of an asymmetric flow field, producing extra aerodynamic force, and changing the flying attitude. Based on the lateral jet principle, this paper investigates the feasibility of applying the idea of a nosebleed air jet to control flying attitude.

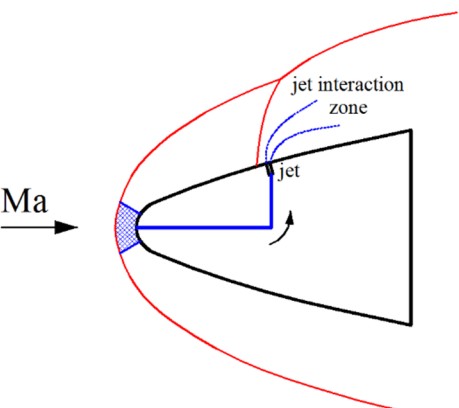

**Figure 1.** Concept of nosebleed air jet.

In hypersonic wind tunnel tests, the total pressure after the shock wave is often measured with a pitot tube, and the total pressure before the shock wave is added to determine the incoming flow Mach number. Assuming the detached shock wave corresponding to the stagnation zone is a normal shock wave, the fluid parameters in the stagnation zone of the blunt cone aircraft are primarily determined by the flight Mach number. The relationship between the flow parameters before and after the normal shock wave is deduced and analyzed by using the basic equations of the one-dimensional steady flow theory of

ideal gas. Based on the principle of pitot tube measurement parameters, the ratio of the stagnation pressure following the shock wave $(P_2{}^*)$ to the incoming flow static pressure $(p_0)$ can be calculated, as shown in Formula (1).

$$\frac{P_2{}^*}{p_0} = \left(\frac{\gamma+1}{2}\right) Ma^2 \left(\frac{\frac{\gamma+1}{2}Ma^2}{\frac{2\gamma}{\gamma+1}Ma^2 - \frac{\gamma-1}{\gamma+1}}\right)^{\frac{1}{\gamma-1}} \tag{1}$$

Taking a flight height of 25 Km ($p_0$ = 2549 Pa, $t_0$ = 221.6 K) as an example, Figure 2 exhibits the relationship between airflow pressure in the blunt cone aircraft's stagnation zone and Mach number. The graphic shows that the pressure in the head stagnation zone steadily rises as the Mach number increases. Although the compression of the airflow by the normal shock wave causes significant flow loss, the pressure in the head stagnation zone at $Ma$ = 5 is 32.7 times that of the incoming flow static pressure. At $Ma$ = 6, the head stagnation zone pressure is 46.8 times the incoming flow static pressure, which is 43.1% higher than the value at $Ma$ = 5. It is feasible to utilize the high-pressure gas in the head stagnation zone as the jet's driving gas source without any dependence on airflow pressure.

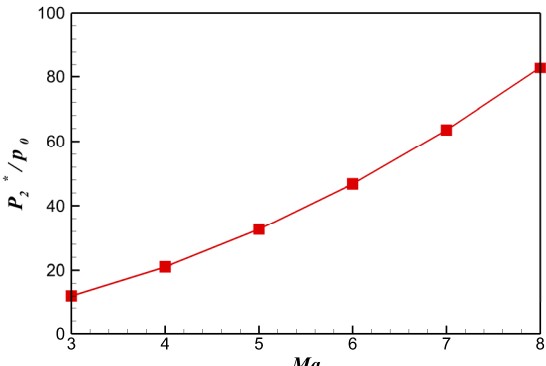

**Figure 2.** The variation of blunt cone model stagnation zone pressure with incoming flow Mach number.

As shown in the graphic, the flow velocity in the stagnation zone is rather low and placed behind the estimated normal shock wave. When establishing the jet channel, the convergent–divergent nozzle (Laval) design theory is employed to optimize the jet velocity as much as possible. As a result, a throat must be installed between the bleed air channel and the jet channel. The high-pressure gas is compressed to sound velocity through the throat before being propelled through the nozzle to eject out at a higher Mach number and interact with the flow surrounding the aircraft.

### 2.2. Configuration of Computational Model

Based on the German Aerospace Research Center (DLR) lateral jet wind tunnel test model with the classical components of the cone, cylindrical fuselage, and flare [16], a model with a blunt cone is constructed. The Haack line is used for the blunt head generatrix. The nose cone is built with a blunt head of $R$ = 10 mm to block airflow at the head and generate a high-pressure zone. As a result, the model head has an aspect ratio of 3.2. Moreover, the lengths of the cylindrical body and skirt tail are appropriately adjusted. The two-dimensional schematic of the blunt cone model is shown in Figure 3. The specific geometric parameters are as follows: the cylindrical body has a diameter of $D$ = 156 mm and a length of 1.56$D$. The skirt tail has a 10° expansion angle and a length of 1$D$. The model's entire length is $L$ = 898.8 mm.

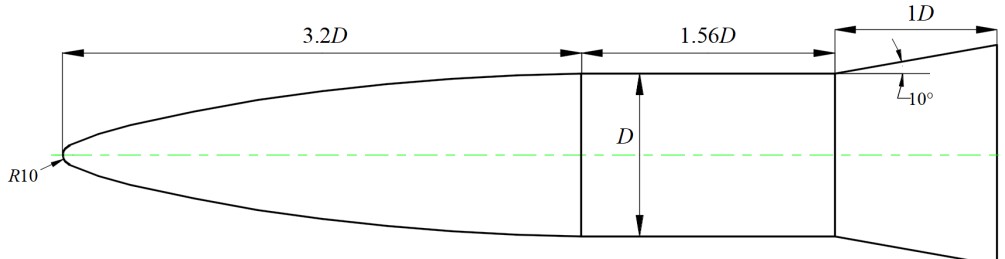

**Figure 3.** Two-dimensional scheme of blunt cone model.

The bleed air flow channel from the stagnation zone is equivalent to a high-pressure cavity, which may be easily built using a straight pipe. The jet channel is intended to be an expansion nozzle. Figure 4 illustrates the two-dimensional schematic with a nosebleed air jet. The inner flow channel of the nosebleed air jet has five key design parameters: flow channel diameter of the bleed air ($A$), throat diameter ($h_{throat}$), jet position ($x_{jet}$), nozzle expansion angle ($\beta$), and jet angle ($\delta$). To obtain appropriate design parameters, $A$ = 12 mm and $h_{throat}$ = 6 mm was chosen, and the investigation was carried out at various $x_{jet}$, $\beta$, and $\delta$. The inner flow channel parameters of the three-dimensional model are determined as $\delta$ = 90°, $\beta$ = 12°, and $x_{jet}$ = 1D by comparing the influence of the nosebleed air jet on the flow field and aerodynamic performance.

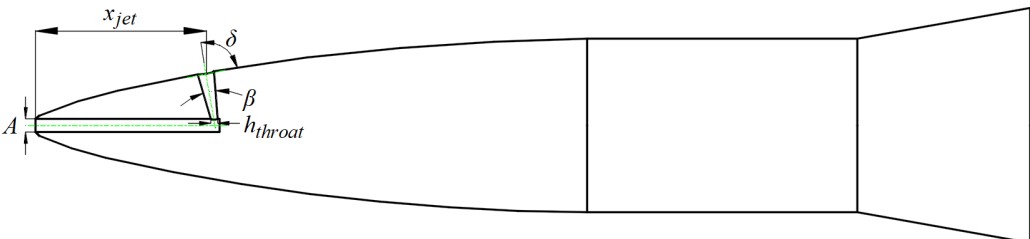

**Figure 4.** Two-dimensional scheme of blunt cone model with nosebleed air jet.

The aerodynamic configuration of the nosebleed air jet's whole inner flow channel may be established, and the configuration scheme can be derived, as shown in Figure 5. The green surface in the graphic represents the inner flow channel's aerodynamic design. The nozzle is located at $\varphi$ = 0°. In other words, when the angle of attack ($AOA$) is positive ($AOA > 0$), the nozzle is located on the leeward side. When the angle of attack is negative ($AOA < 0$), the nozzle is located on the windward side.

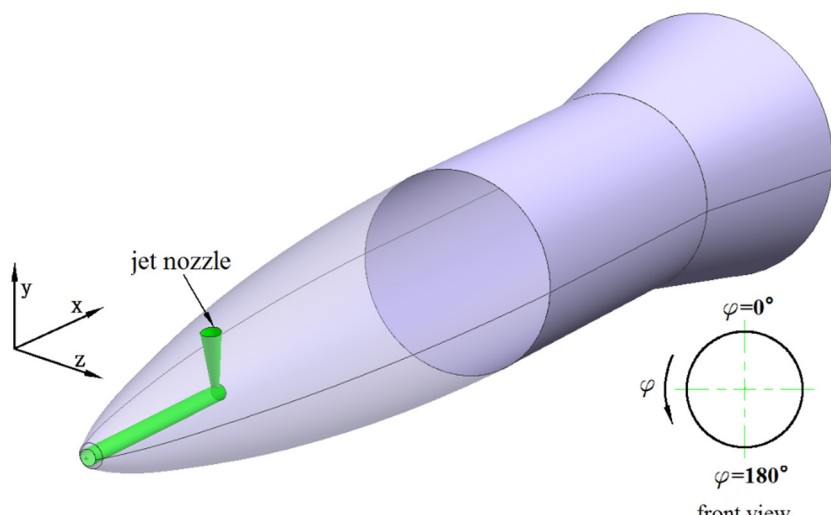

**Figure 5.** Configuration scheme of blunt cone model with nosebleed air jet.

### 2.3. Computational Fluid Dynamics Method and Verification

In this paper, the flow field and aerodynamic performance of the nosebleed air jet are calculated using commercial software (ANSYS Fluent) under hypersonic flow conditions, which solves three-dimensional conservative Reynolds-averaged Navier–Stokes equations (RANS). In the Cartesian coordinates, the governing equations can be transformed into integral form as follows:

$$\frac{\partial}{\partial t} \iiint_V Q dV + \oiint_{\partial V} F \cdot n dS = 0 \tag{2}$$

where $V$ is the control volume, $\partial V$ is the boundary of the control volume, $Q = (\rho, \rho u, \rho v, \rho w, \rho e)^T$ are the conserved variable vectors, $F$ is the vector flux through the control volume surface, including the convection term $F_c$ and the viscosity term $F_v$, and $n$ is the normal vector outside the boundary of the control volume.

The finite volume method is employed to achieve time marching while solving the governing equations. The convection term is discretized by the second-order Roe scheme, the viscosity term is discretized by the central difference scheme, and the time term is discretized by the lower-upper symmetric Gauss–Seidel (LU-SGS) implicit scheme. All walls of the model are treated as adiabatic. To capture the near-wall region flow, standard wall functions are added, and molecular viscosity coefficients are calculated using Sutherland's law.

Considering the high stagnation temperature, the variable specific heat is adopted, which is a polynomial function of temperature, and its coefficients are derived from the Joint Army, Navy, and Air Force (JANAF) thermodynamic table. In the numerical solutions, the boundary conditions (i.e., symmetry, pressure far field, pressure outlet, and wall boundaries) are adopted. According to the numerical calculation method in references [31–35], the two-equation eddy viscosity $k$-$\omega$ SST turbulence model is adopted to model the internal and external turbulent flow of the nosebleed air jet, which can accurately and precisely describe the problem studied. The convergence of each case is determined by the residual history of each governing equation and the mass flow rate of the jet. When the residual of each equation drops by four orders of magnitude and the mass flow rate of the jet is stable, the calculation is considered converged.

To verify the above-mentioned numerical calculation method, the DLR's cone-cylindrical fuselage-flare model is utilized as the object, and its aerodynamic configuration is illustrated in Figure 6. The cylindrical fuselage has a diameter of $D = 40$ mm. The cone angle is 40°, and $4.3D$ downstream from the nose, there is one side jet with a diameter of $0.1D$. At the exit, there is a cylindrical nozzle with $Ma_{jet} = 1$. The lateral jet nozzle is positioned at an azimuth angle of $\varphi = 180°$. The numerical simulation results obtained using the calculation method in this paper are compared with the experimental results of $Ma = 3$ in reference [16]. The experimental conditions are as follows: the incoming Mach number is $Ma = 3$, the angle of attack is 0°, the jet gas is air, the jet pressure ratio (relative to the incoming static pressure) is 200, the total temperature of the jet gas is 280 K, and the jet gas Mach number is $Ma_{jet} = 1.0$.

The flow field is numerically simulated using the above-mentioned numerical calculation method under the experimental condition of $Ma = 3$. Figure 7 compares the pressure distribution along the wall between the numerical simulation and experimental data with $\varphi = 180°$, 150°, 120° and 90°. The $y$-coordinate in the figure indicates the difference in the wall pressure coefficient with jet and without jet, provided by $(C_P)_{diff}$, $(C_P)_{diff} = (C_P)_{with\ jet} - (C_P)_{without\ jet}$, where $C_P = 2(P - P_0)/(P_0 \gamma Ma_0^2)$. The $x$-coordinate represents the model's size in the $x$ direction, which is dimensionless considering the cylindrical fuselage's diameter $D$.

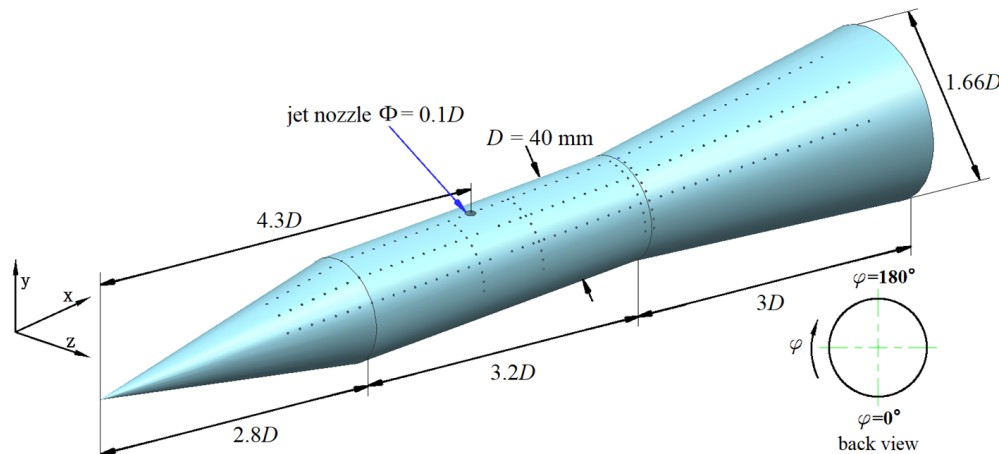

**Figure 6.** Typical lateral jet test model obtained according to reference [16].

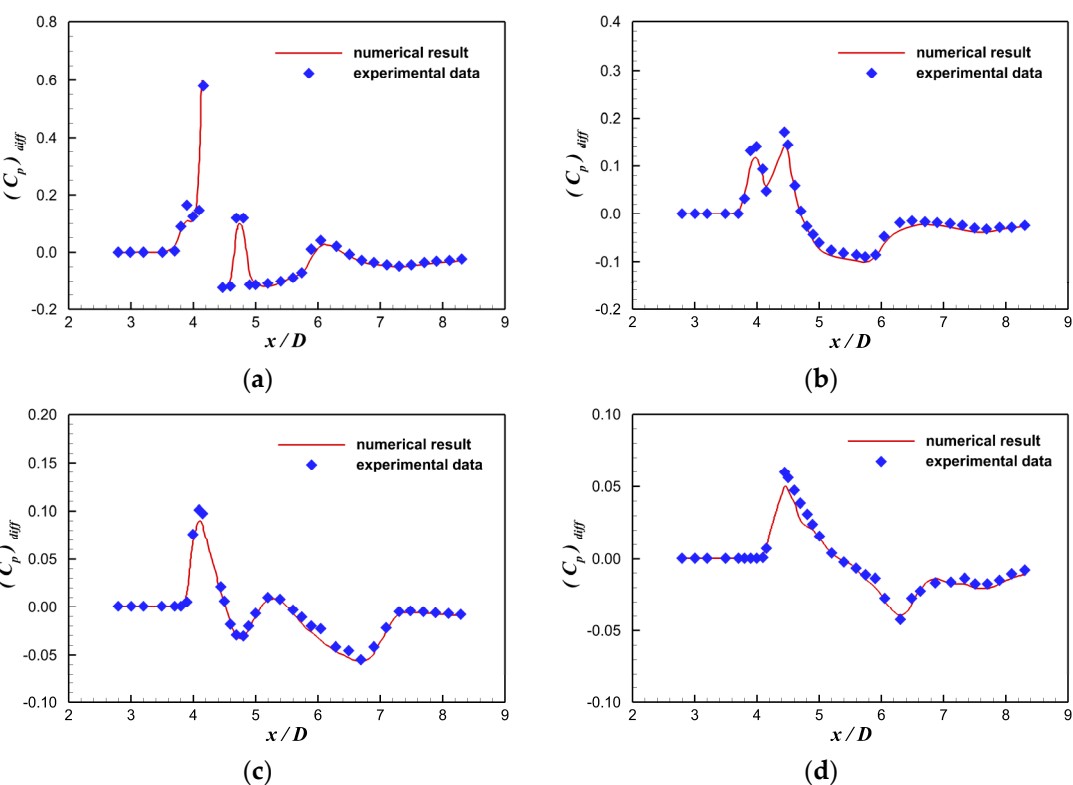

**Figure 7.** Comparison of pressure distribution along the wall between numerical simulations and experiments. (**a**) $\varphi = 180°$; (**b**) $\varphi = 150°$; (**c**) $\varphi = 120°$; (**d**) $\varphi = 90°$.

The figure shows that the calculation method in this paper can accurately model the changing trend of pressure distribution along the wall, especially the position of the pressure jump, which agrees well with the experimental data. However, there are some differences between the numerical simulation results and the experimental data near the peak point of wall pressure. The interaction between the jet and the flow surrounding the aircraft causes extremely complicated spatial shock waves. Meanwhile, the jet and the associated shock waves will cause significant boundary layer separation, resulting in a reattachment shock wave. Jet interaction will cause severe vortex flow in the flow field, which will inevitably bring great difficulties to the numerical simulation. In general, the numerical calculation method used in this paper is capable of predicting the complicated flow caused by the interaction of the jet with the flow surrounding the aircraft, and the calculation results are credible.

## 3. Numerical Investigation of the Nosebleed Air Jet in a Blunt Cone Aircraft

### 3.1. Treatment of Numerical Accuracy

According to the aerodynamic configuration (as shown in Figure 5), the grid is generated using ANSYS ICEM. Figure 8 exhibits the computational grids around the model and on the wall surface. Grid topology and the grid density are adjusted in consideration of the jet and supersonic crossflow surrounding the vehicle interaction structure. The grid system includes the inlet and jet nozzle parts to account for the boundary layer effects of the entire inner flow channel. The first wall cell spacing at the surface is set as $1 \times 10^{-6}$ m from the model wall surface in order to satisfy the dimensionless wall distance ($y^+$) value near 1 for the resolution of turbulent flows. Table 1 lists the boundary conditions of numerical computation.

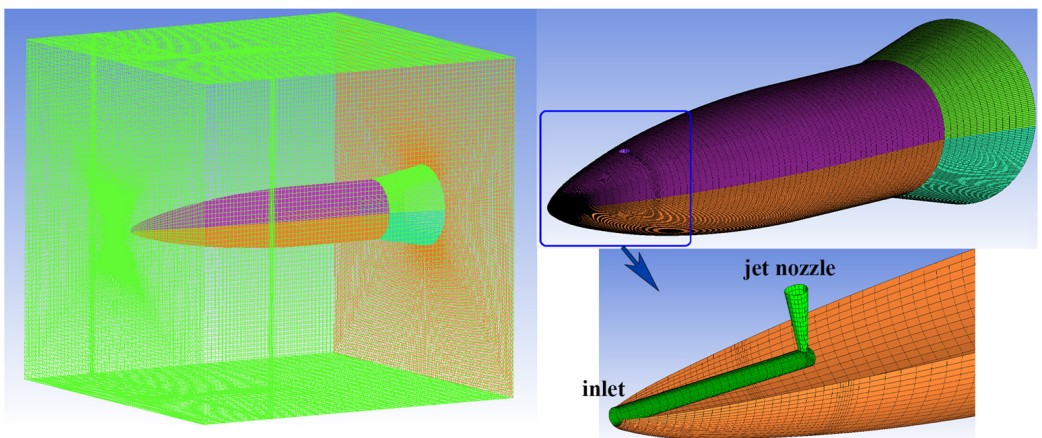

**Figure 8.** Computational domain and grids.

**Table 1.** Boundary conditions.

| Property | Value |
|---|---|
| Mach number | 5 |
| Total pressure, MPa | 1.0 |
| Total temperature, K | 360 |

A proper grid dependency study was performed before the final grid selection. Three progressively finer grids consisting of 1.65 million (coarse), 3.42 million (medium), and 5.36 million (fine) hexahedral cells are employed for the grid refinement test. For each grid size, the grid topology is the same. Table 2 lists the comparison of computational results with the experimental value. It is evident that the differences between the medium grid and the fine grid are 1.37% at $AOA = 6°$, 1.85% at $AOA = 0°$, and 2.12% at $AOA = -6°$ for the distance from the shock to the surface at the nozzle position of x = 1D. Overall, the differences in the medium grid under different angles of attack are nearly 5% compared with those of the experimental values.

**Table 2.** Grid sensitivity analysis for the nosebleed air jet tests.

| Grid Case | No. of Cells (Million) | Distance between the Shock and the Surface at the Nozzle Position of *x* = 1D, mm | | |
|---|---|---|---|---|
| | | *AOA* = 6° | *AOA* = 0° | *AOA* = −6° |
| Coarse | 1.65 | 32.6 | 29.1 | 24.5 |
| Medium | 3.42 | 34.7 | 30.8 | 26.7 |
| Fine | 5.36 | 35.2 | 31.4 | 27.3 |
| Experimental value | – | 36.4 | 32.5 | 28.3 |

In addition, the grid convergence index (GCI), as described by Roache [36], is also computed, taking the suggested factor of safety of 1.25 for a three-grid study. In this paper, the wall surface pressure has been taken as the parameter of interest. For the pressure, the GCI is seen to be less than 10%. Thus, the computational results may be considered to be grid independent. Furthermore, the variation of solutions when going from the medium grid to the fine grid is less significant compared with that from the coarse grid to the medium grid. Therefore, the medium grid consisting of 3.42 million cells is chosen as a reasonable tradeoff between the numerical accuracy and solution time.

### 3.2. Influence on the Flow Field

Consider two typical angles of attack ($AOA$ = 6° represents that the nozzle is located on the leeward side, and $AOA$ = −6° represents that the nozzle is located on the windward side). Figure 9 illustrates the flow field generated by the interaction of the jet with the flow surrounding the aircraft, as well as the local flow around the nozzle. The jet nozzle is shown in Figure 5 to be at a position of $\varphi$ = 0° and $x_{jet}$ = 1D. When the incoming flow has a negative angle of attack, the environmental pressure on the aircraft's surface near the nozzle is quite high. The nozzle's exit will be subjected to strong back pressure, which will significantly hinder the jet. The maximum velocity in the nozzle is $Ma$ = 2.41 at $AOA$ = 0°, based on the local flow around the nozzle. Due to the back pressure, there is severe flow separation in the nozzle, and the velocity of the outflow mainstream decreases to $Ma$ = 1.25. Furthermore, the interaction between the jet and the flow surrounding the aircraft dramatically alters the flow on the upper wall. The thickening of the boundary layer, in particular, corresponds to a significant change in the upper wall configuration.

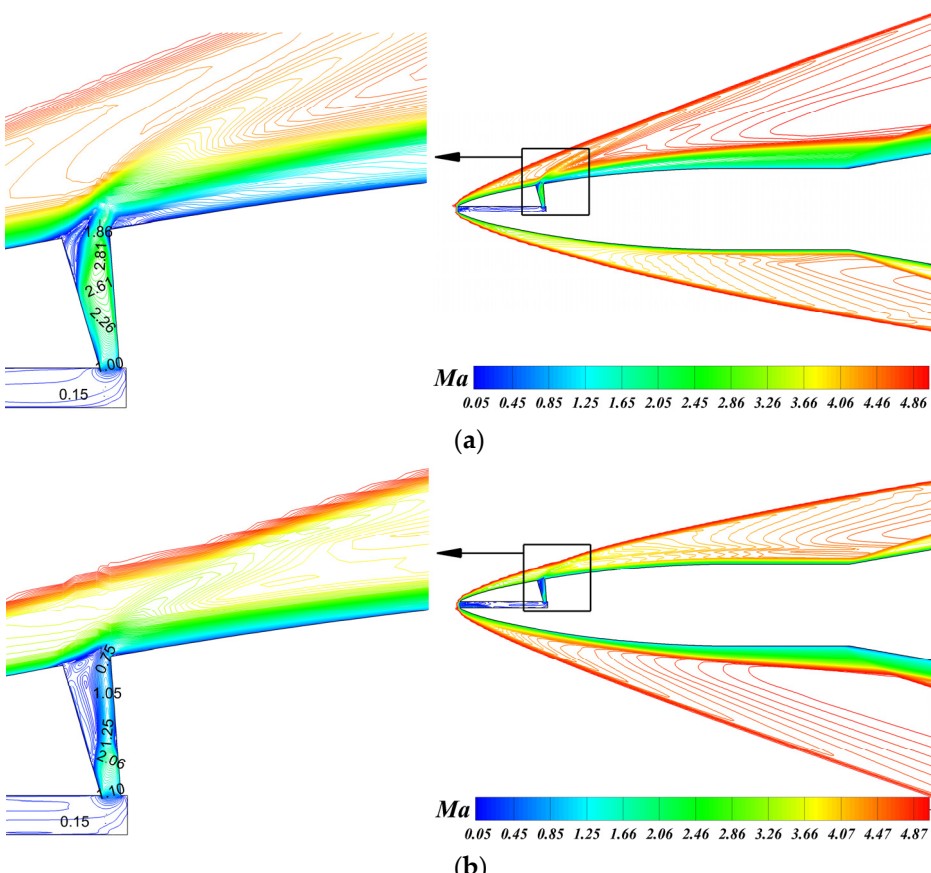

**Figure 9.** Flow fields of $Z$ = 0 section at $Ma$ = 5 and different angles of attack. (**a**) $AOA$ = 6°; (**b**) $AOA$ = −6°.

Six sections of $X$ = 145 mm, 156 mm, 170 mm, 200 mm, 230 mm, and 260 mm were intercepted before and after the nozzle position to demonstrate the flow variations generated by the interaction between the jet and the flow surrounding the aircraft along the flow direction. Figure 10 illustrates the flow fields of several $X$ sections near the nozzle at $Ma$ = 5 and different angles of attack, as well as the change in the boundary layer generated by the jet. The figure shows that when $AOA$ > 0, the nozzle is located on the leeward side, and the wall boundary layer is thick. As a result of the jet, there is significant separation, and an obvious vortex forms along the flow direction (as shown in Figure 10a). There is essentially no vortex along the flow direction when the nozzle is located on the windward side ($AOA$ < 0).

Figure 11 illustrates the wall pressure distributions of several circumferential sections at $Ma$ = 5 using $AOA$ = 6°, 0°, and −6° as examples. It can be seen that the separation shock wave generated by the jet grows more than 30° in the circumferential direction, and the pressure distribution exhibits the characteristic of pressurization caused by the shock wave in the range of $\varphi$ = 0°~30°. Because the section of $\varphi$ = 45° is far away from the nozzle exit position, the separation shock wave has no direct influence on the wall pressure distribution. When the value of $\varphi$ is small, the pressure distribution's change trend in the circumferential section is consistent with that of $\varphi$ = 0°. However, the intensity of the separation shock wave generated by the jet gradually lessens along the circumferential direction. As a result, the pressurization impact on the flow will gradually decrease, as seen by a steady decline in the pressure peak caused by the shock wave. Furthermore, the pressure distributions of different $\varphi$ sections are identical before reaching the jet position at $AOA$ = 0°. Due to the boundary layer asymmetry at $AOA$ = 6° and −6° (as shown in Figure 10a,e), the pressure distribution of $\varphi$ = 45° section is separate from the other curves before reaching the jet position.

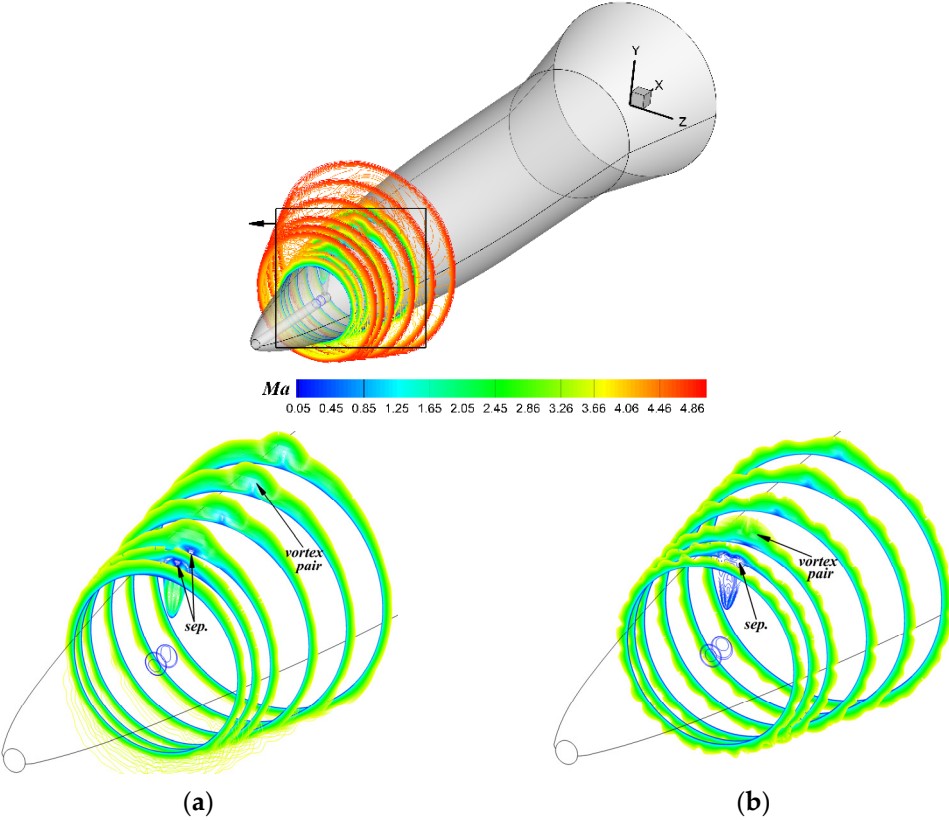

(**a**)            (**b**)

**Figure 10.** *Cont.*

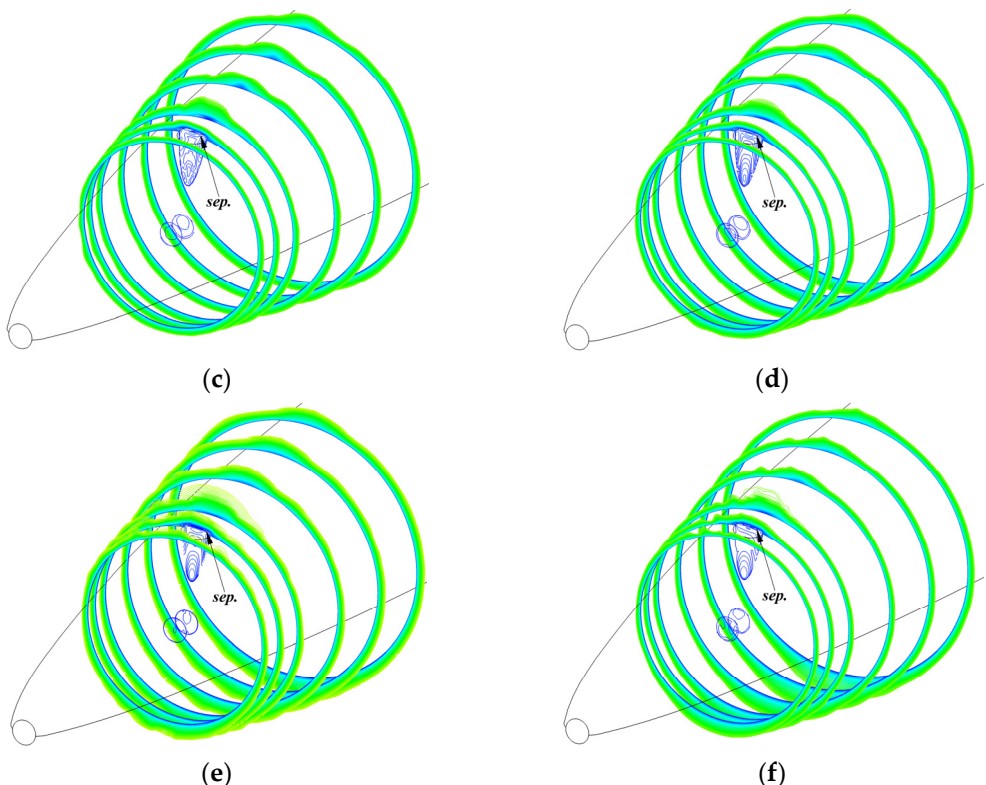

**Figure 10.** Flow fields of several *X* sections near the nozzle at *Ma* = 5 and different angles of attack. (**a**) *AOA* = 6°; (**b**) *AOA* = 0°; (**c**) *AOA* = −2°; (**d**) *AOA* = −4°; (**e**) *AOA* = −6°; (**f**) *AOA* = −8°.

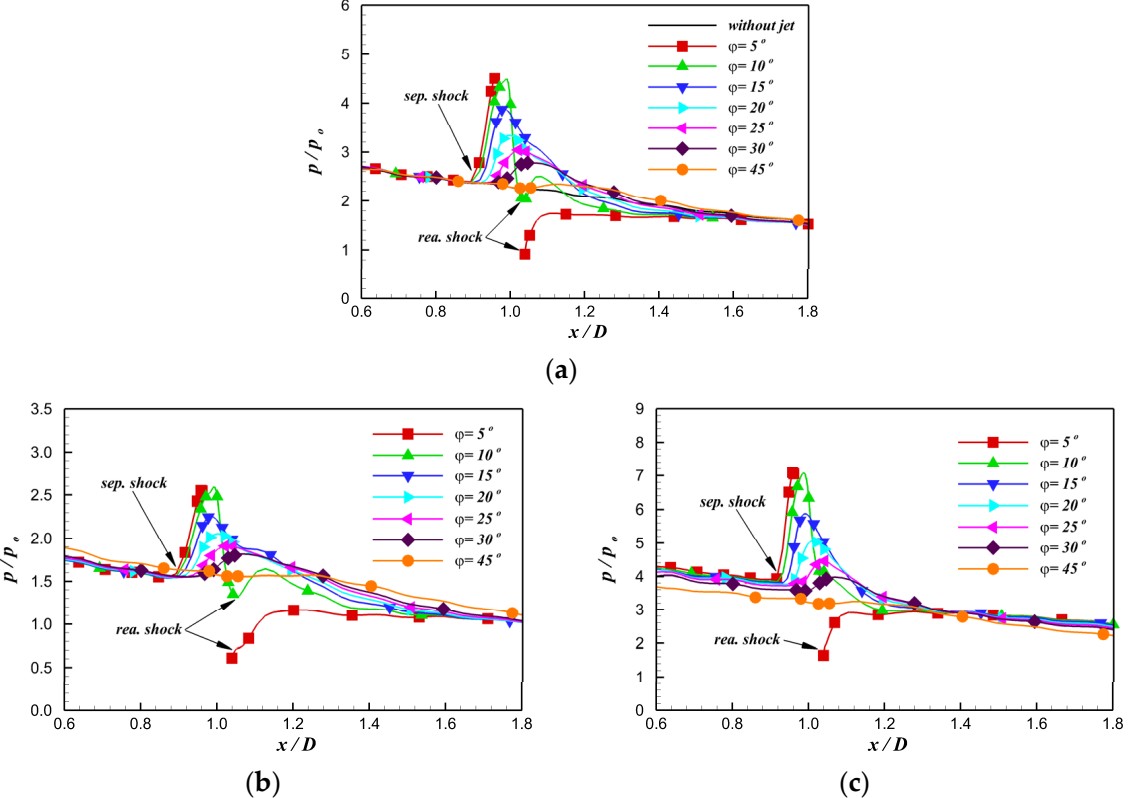

**Figure 11.** Wall pressure distribution comparison of several *φ* sections at *Ma* = 5. (**a**) *AOA* = 0°; (**b**) *AOA* = 6°; (**c**) *AOA* = −6°.

Taking the two sections of $\varphi = 10°$ and $30°$ as examples, Figure 12 compares their wall pressure distributions near the nozzle exit position at different angles of attack. The separation shock wave generated by the jet develops more than $30°$ in the circumferential direction, as seen in the diagram. The pressure distribution on the section of $\varphi = 30°$ has been pressurized as a result of the shock wave. The pressurization of the separation shock wave is the weakest at $AOA = 6°$. The intensity of the separation shock wave steadily increases as the angle of attack decreases. For the pressure distribution of the $\varphi = 10°$ section (near the nozzle exit), when $AOA = 6°$ and $0°$, the boundary layer near the nozzle thickens (as shown in Figure 10), and an obvious reattachment shock wave forms downstream of the nozzle, considerably increasing the pressure. When $AOA < 0$, the boundary layer near the nozzle grows thinner and thinner owing to the downwash of the incoming flow and the strong back pressure, and the separation zone generated by the jet gradually decreases, and no reattachment shock wave forms downstream of the nozzle. At the angle of attack of $-6°$, for example, there is essentially no reattached shock wave downstream of the nozzle, and the pressure distribution does not appreciably change.

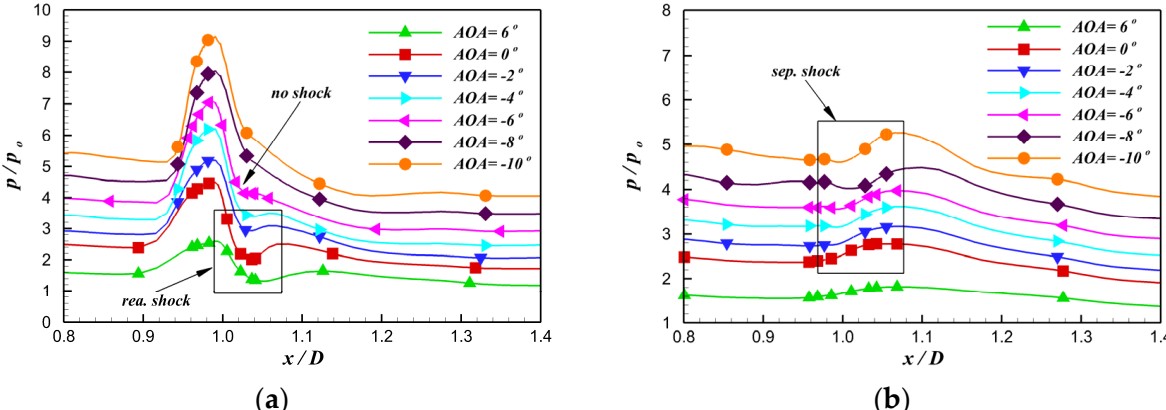

**Figure 12.** Comparison of wall pressure distribution near the nozzle exit position at different angles of attack. (**a**) $\varphi = 10°$; (**b**) $\varphi = 30°$.

Figure 13 illustrates the wall streamlines near the nozzle at three different angles of attack when $Ma = 5$. The separation line generated by the jet is very close to the nozzle in front of the nozzle, as shown in the figure, due to the relatively low flow pressure ratio and flow rate when the jet is introduced from the nose. At $AOA = 6°$, the jet interaction is significant, and the flow fields such as the separation line and reattachment point near the nozzle are plainly evident. Due to the incoming flow, the separation lines on both sides of the nozzle downstream will be close to the axis. The separation zone around the nozzle steadily decreases as the angle of attack lowers, and the separation line ultimately extends to both sides of the nozzle.

*3.3. Influence on the Aerodynamic Performance*

Figure 14 illustrates the variation of average Mach number and average pressure at the nozzle exit with angle of attack. According to the graphic, when the nozzle is located on the leeward side of the aircraft (at a positive angle of attack), the environmental pressure on the aircraft surface is low, resulting in low back pressure on the nozzle. As a result, the flow pressure ratio is lowest near the nozzle exit and the Mach number is the highest. At $AOA = 6°$, for example, the average Mach number at the nozzle exit is close to 1.6. When the nozzle is located on the windward side (the angle of attack is negative), the environmental pressure gradually rises as the angle of attack increases. Therefore, the nozzle's back pressure continues to rise, as seen by an increasing flow pressure ratio and a decreasing Mach number at the nozzle exit. Due to the significant back pressure, severe separation and congestion develop in the nozzle, which is equivalent to flow stagnation (the flow at the aircraft's nose has previously been stagnant). Furthermore,

in the calculation stage, the air density is low, and the cross-sectional area of the bleed channel is fixed (its diameter is 12 mm). Thus, the inner channel's mass flow rate is extremely low. In this way, the jet generates very little thrust. Therefore, the interaction between the jet and the flow surrounding the aircraft has the greatest influence on the aircraft's aerodynamic performance.

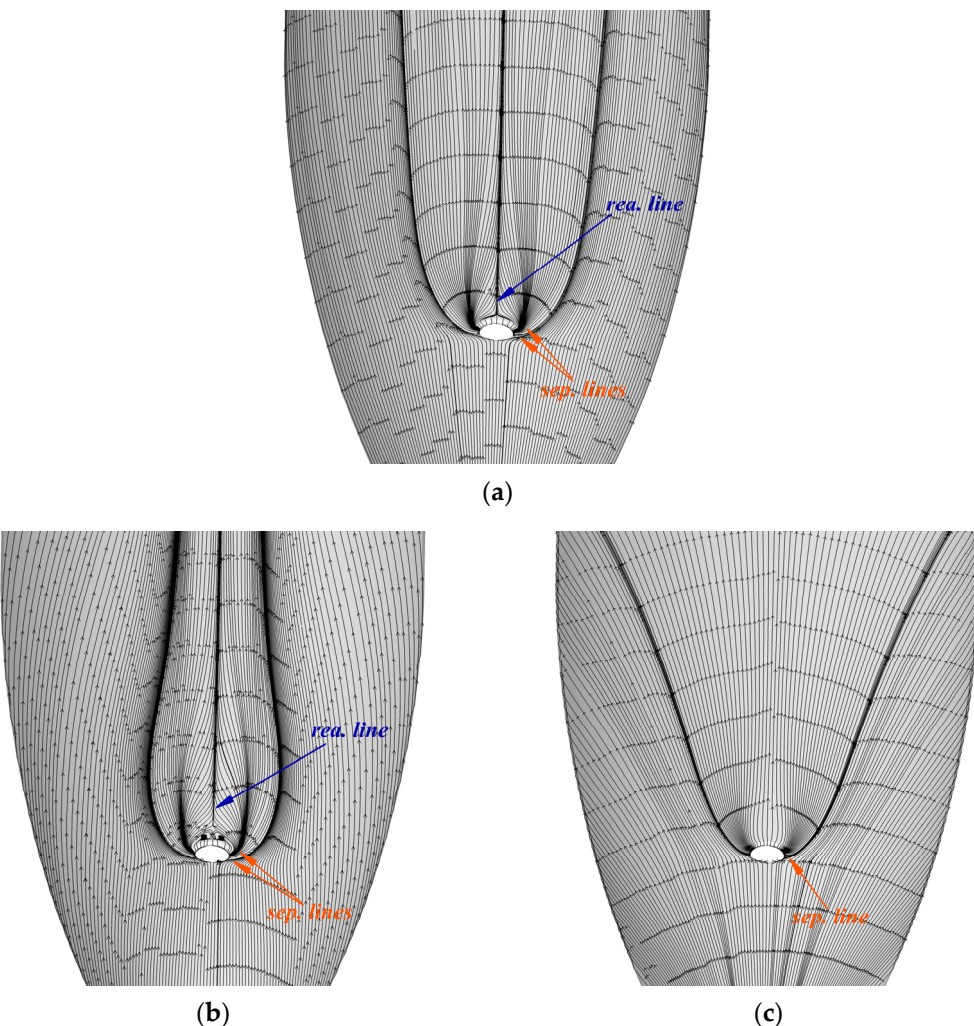

**Figure 13.** Wall streamlines near the nozzle at *Ma* = 5 and different angles of attack. (**a**) *AOA* = 0°; (**b**) *AOA* = 6°; (**c**) *AOA* = −6°.

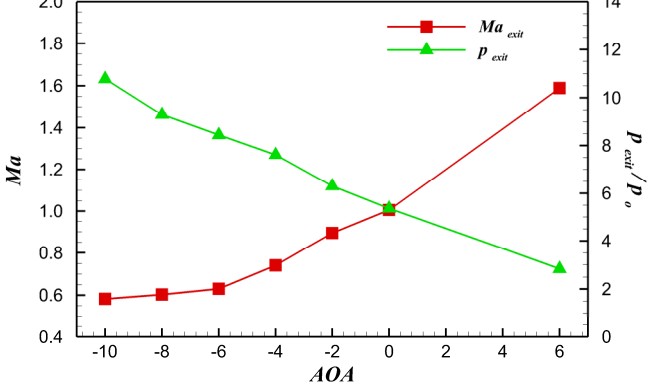

**Figure 14.** The relationship between Mach number and pressure at the inner channel exit with the angle of attack.

The nosebleed air jet alters the aircraft's local flow field and pressure distribution, which has a direct influence on the aerodynamic force. The major purpose of this study is to compare aerodynamic performance parameters such as normal force, pitching moment, axial force coefficient, center-of-pressure coefficient, pitching moment coefficient, etc. Table 3 exhibits the normal force generated by the jet interaction at *Ma* = 5 and different angles of attack. The normal force without the jet is also provided in the table for comparison. The nosebleed air jet differs from the traditional lateral jet that has an attitude control motor. The diameter of the bleed channel is 12 mm, and the mass flow rate in the inner channel is quite low. The jet creates relatively little thrust in this way. Jet interaction, rather than the jet itself, generates most of the aerodynamic force. According to the table, the jet causes the model's normal force to change. At *AOA* = +6°, the normal force with the jet is 0.6% lower than that without the jet. However, at *AOA* = −6°, the normal force with the jet decreases by 0.2%.

**Table 3.** Comparison of normal force between with and without jet at *Ma* = 5.

| AOA | +6° | 0° | −2° | −4° | −6° | −8° | −10° |
|---|---|---|---|---|---|---|---|
| Without jet | 520.4 | 0.1 | −167.5 | −338.0 | −520.4 | −720.3 | −938.2 |
| With jet | 517.1 | −1.2 | −168.2 | −338.0 | −519.3 | −717.7 | −937.6 |

Figures 15 and 16 illustrate the axial force coefficient and center-of-pressure coefficient generated by the jet at *Ma* = 5. The values without the jet are also shown in the figure for comparison. According to Figure 15, the jet substantially raises the axial force coefficient at *AOA* = −2° and −4°. Compared with the case without the jet, the axial force coefficient with the jet increases by 1.3% at *AOA* = −2°, and by 1.2% at *AOA* = −4°. Furthermore, the interaction between the jet and the flow surrounding the aircraft alters the wall flow (as shown in Figure 16). As a result, the jet primarily affects the aircraft's center-of-pressure coefficient, especially when the angle of attack is low (−4° to 0°).

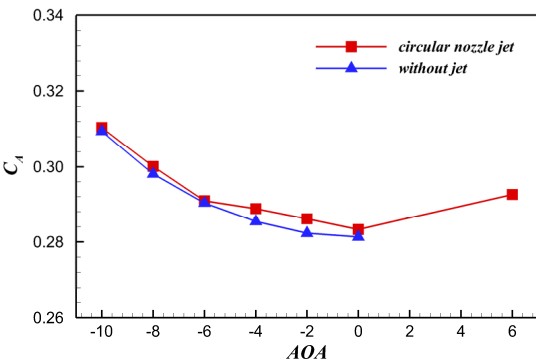

**Figure 15.** The variation of axial force coefficient with angle of attack.

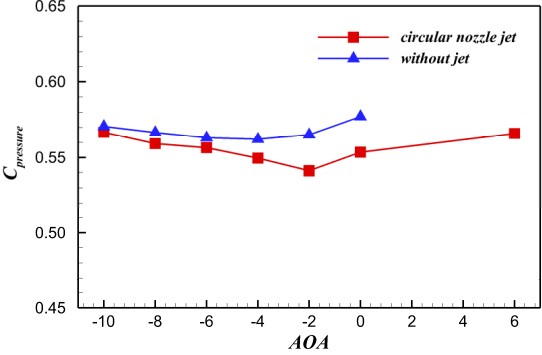

**Figure 16.** The variation of center-of-pressure coefficient with angle of attack.

The pitch moment may be calculated after determining the normal force, the pressure center position, and the torque reference center. To calculate the pitching moment coefficient, choose (0.5 *L*, 0, 0) as the moment reference center, the reference length as the model's axial length, and the reference area as the cross-sectional area of the cylindrical section. Figure 17 illustrates the variation of the pitching moment coefficient with the angle of attack. According to the graphic, at *AOA* = −4°, the pitching moment coefficient without the jet is 0.01819, and that with the jet is 0.01436. The jet reduces the pitching moment coefficient by 21.1%. The jet causes a significant difference in the pitching moment coefficient when compared with the case without the jet. For the nosebleed air jet, the difference in aerodynamic performance between with and without the jet may be improved further by increasing the mass flow rate of the inner channel or modifying the jet nozzle.

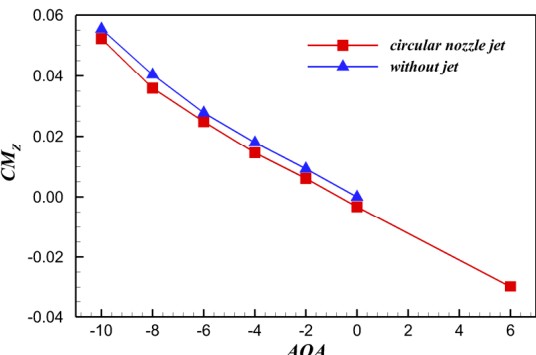

**Figure 17.** Pitch moment coefficient generated by the jet.

## 4. Wind Tunnel Test Demonstration of the Nosebleed Air Jet

### 4.1. Experimental Model

According to the aerodynamic configuration of the nosebleed air jet shown in Figure 5, the experimental model is developed with a suitable size based on the Φ0.5 m hypersonic wind tunnel (FL-31) of the China Aerodynamics Research and Development Center (CARDC). Figure 18 illustrates the experimental model's aerodynamic configuration. The circular nozzle's throat has a diameter of Φ = 3 mm and an expansion angle of *β* = 12°. The jet is angled at 90° to the aircraft's wall surface.

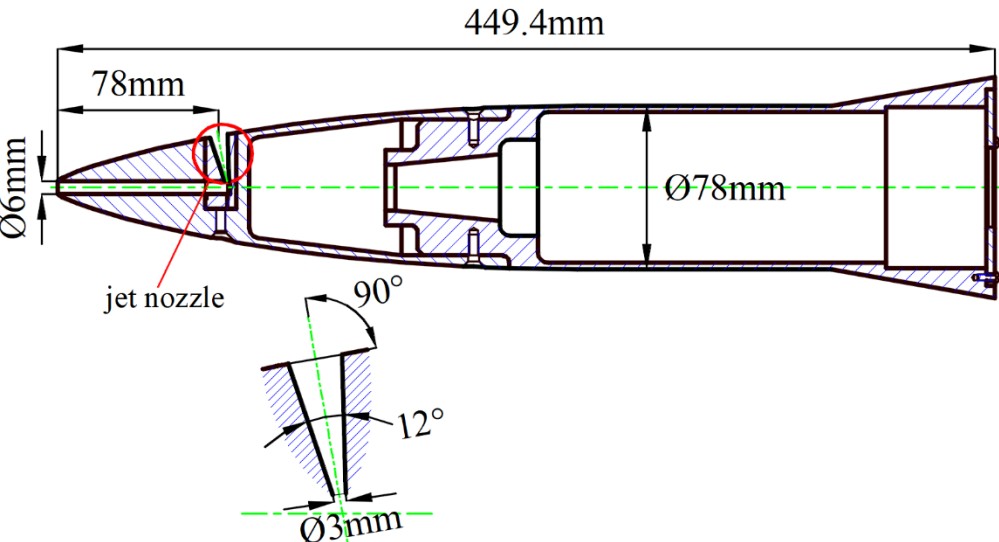

**Figure 18.** Aerodynamic configuration of experimental model.

Figure 19 illustrates the three-dimensional model, as well as the installation position of the balance. The actual model is shown in Figure 20. To close the bleed air channel, insert the throttling cylinder into the inlet channel while leaving the jet channel alone. Figure 21

exhibits the photographs of the nosebleed air jet test model installed in the wind tunnel. The aerodynamic force is measured using a balance arranged inside the model, and the experimental flow field is studied using the schlieren system.

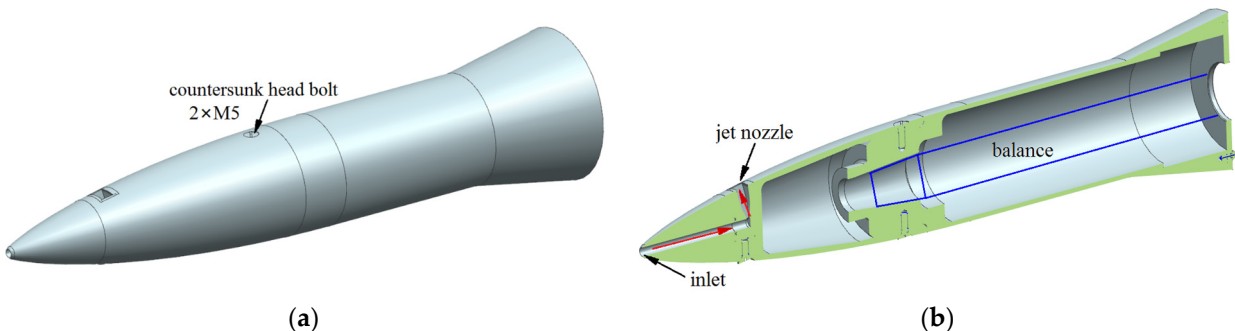

(**a**)

(**b**)

**Figure 19.** Three-dimensional model of blunt cone with nosebleed air jet. (**a**) three-dimensional model; (**b**) profile.

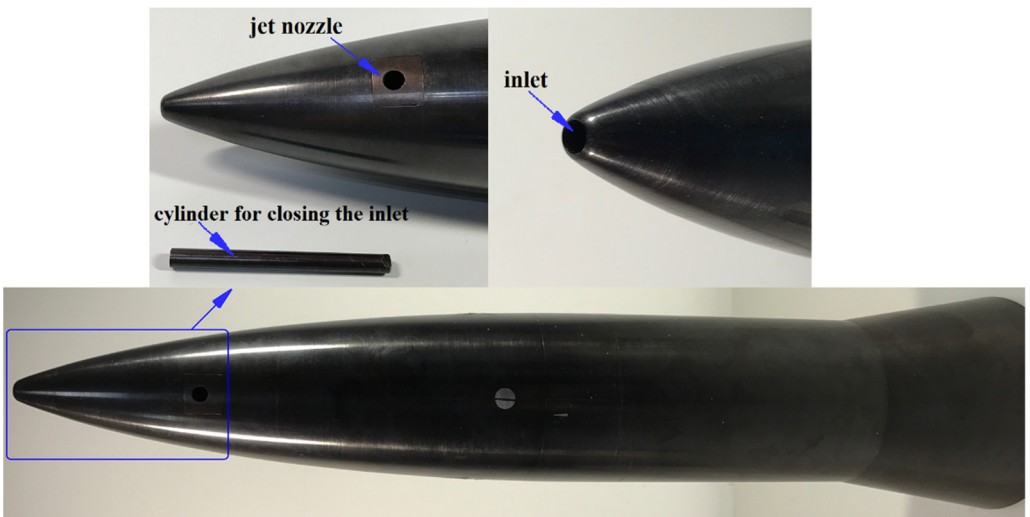

**Figure 20.** Photographs of the experimental model.

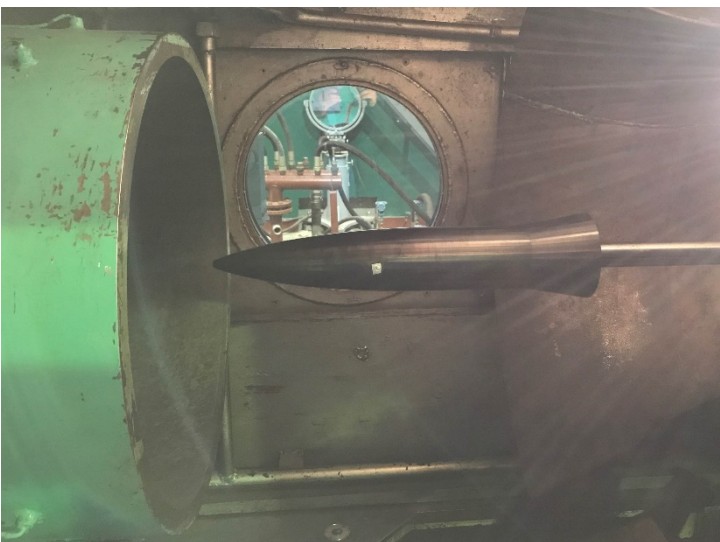

**Figure 21.** Photographs of the experimental model installed in the wind tunnel (FL-31).

### 4.2. Wind Tunnel and Measurement System

The experiments were performed at the Φ0.5 m hypersonic wind tunnel (FL-31) of the China Aerodynamic Research and Development Center (CARDC) [37]. The FL-31 is an intermittent blow-down direct heating conventional hypersonic wind tunnel. It primarily consists of a high-pressure air supply, a heater, a nozzle and test section, a diffuser, a vacuum system, an electrical control system, a data acquisition and processing system, a Φ300 mm color schlieren system, etc. The preheating air (anti-condensation) was used as the working medium. The heater's total power reaches 8350 kW. The facility provides nominal freestream Mach numbers from 5.0 to 10.0 with an exit diameter of 500 mm. The tests can be conducted under the conditions of 0.5 to 11.6 MPa total pressure, 350 to 1073 K plenum temperature, and $0.35 \times 10^7$ to $7.71 \times 10^7$ m$^{-1}$ unit Reynolds number. The usable run time is 60 to 360 s according to the Mach number. The conventional force measuring tests were performed for the jet interaction test under the conditions of *Ma* = 5, with and without the jet. The corresponding wind tunnel flow conditions are summarized in Table 4.

**Table 4.** Flow conditions for the FL-31 wind tunnel test.

| Mach Number | Actual Flow Mach Number | Freestream Stagnation Pressure, MPa | Freestream Static Pressure, Pa | Freestream Stagnation Temperature, K | Freestream Static Temperature, K |
|---|---|---|---|---|---|
| 5 | 4.95 | 1.0 | 2004 | 360 | 61 |

The tail support holds the model to the angle of attack adjusting mechanism. The model is buried in the test section prior to the experiment. After establishing the flow field, the model was delivered to the wind tunnel test diamond region through the adjustment mechanism. The angle of attack in the experiment varied from $-10°$ to $+10°$. The model aerodynamic force was measured with a medium temperature and conventional six-component balance (5N6-24I), which is equipped with an epoxy resin fiberglass insulation device to prevent the temperature effect of the balance element caused by the high temperature of the incoming flow. The strain gauge adopts the SK-MC-062AP-350/SP21 made by VISHAY Company, US. The data acquisition system was performed with the American Agilent company's 34980A-MY53152854 module.

The six-component balance was calibrated before the tests. Table 5 gives the maximum loads and the relative accuracies for each balance component established by the balance calibration. If the calibration resulted in even lower values, the minimum accuracy was set to 0.05% based on previous experience. Furthermore, it is important to note that the accuracy of aerodynamic coefficients is also influenced by the accuracy of flow parameters. The observation and recording of the flow image were completed by the color schlieren system and the computer video digital recording acquisition system. The conventional cyclic scanning data acquisition and processing system was used to collect and process the signals output by pressure sensors, thermocouples, and balances. The cyclic scanning sampling frequency was 100 KHz, and the data acquisition accuracy was better than 0.02%.

**Table 5.** Maximum loads and relative accuracy of six-component measurements.

| Component Cond. | Loads, N/(Nm) | Accuracy, % |
|---|---|---|
| Axial force X | 320 | 0.09 |
| Normal force Y | 960 | 0.15 |
| Side force Z | 320 | 0.25 |
| Rolling moment $M_X$ | 6 | 0.32 |
| Yawing moment $M_Y$ | 12 | 0.30 |
| Pitching moment $M_Z$ | 48 | 0.07 |

### 4.3. Test Results and Discussion

#### 4.3.1. Flow Field Analysis

During the test, the aerodynamic forces are measured at $AOA$ = −10°, −8°, −6°, −4°, −2°, 0°, 2°, 4°, 6°, 8°, and 10°. Figure 22 illustrates the flow field schlieren comparison of with and without the jet at $AOA$ = 0°. It can be seen that the bleed air has little effect on the nose flow field. The interaction between the jet and the flow surrounding the aircraft is obvious, and the bow shock wave gets closer to the model. It is evident from the schlieren comparison that the distance between the shock wave and the wall surface is shortened by 3 mm. Furthermore, the flare generates a symmetrical flow field structure under both jet and without jet conditions. The schlieren photographs are displayed in Figure 23 using $AOA$ = −6° and 6° as examples. The flow field of jet interaction is directly affected by the environmental pressure at the nozzle exit.

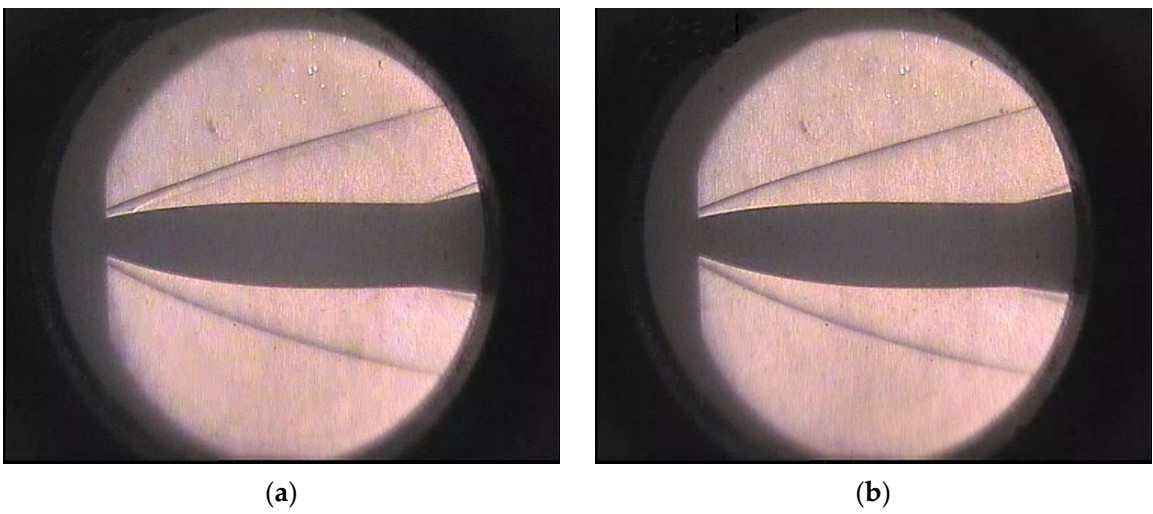

(**a**)　　　　　　　　　　　　　　　　　(**b**)

**Figure 22.** Schlieren photographs of with jet and without jet at $Ma$ = 5 and $AOA$ = 0°. (**a**) with jet; (**b**) without jet.

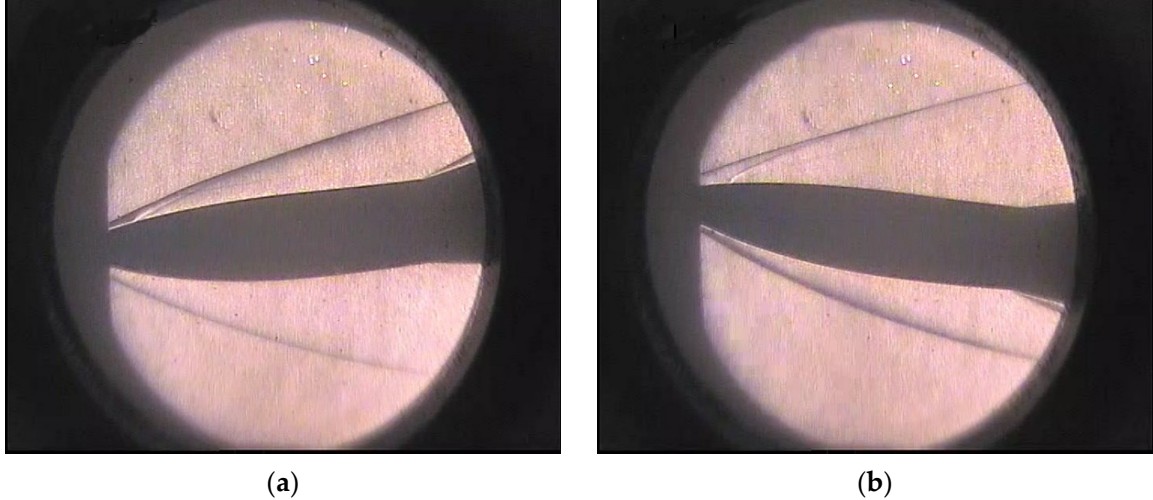

(**a**)　　　　　　　　　　　　　　　　　(**b**)

**Figure 23.** Schlieren comparison of typical angle of attack at $Ma$ = 5. (**a**) $AOA$ = −6°; (**b**) $AOA$ = 6°.

#### 4.3.2. Aerodynamic Performance Analysis

Figure 24 illustrates the variation of the measured normal force coefficient, center-of-pressure coefficient, pitching moment coefficient, and axial force coefficient with angle of attack at $Ma$ = 5. The moment reference center is (0.5 $L$, 0, 0). The numerical simulation results are also provided in the graphic for comparison. It can be seen that the numerical

simulation's aerodynamic performance is essentially consistent with the experimental measurement. The axial force coefficient, on the other hand, greatly varies. The fundamental reason is that the base drag is ignored in the numerical simulation, and the end face of the bottom is directly employed as the flow field outlet. The figure simply provides a reference. In addition, there is a significant center-of-pressure coefficient difference between the numerical and experimental results at $AOA = 2°$ (as shown in Figure 24b) due to the unstable separation.

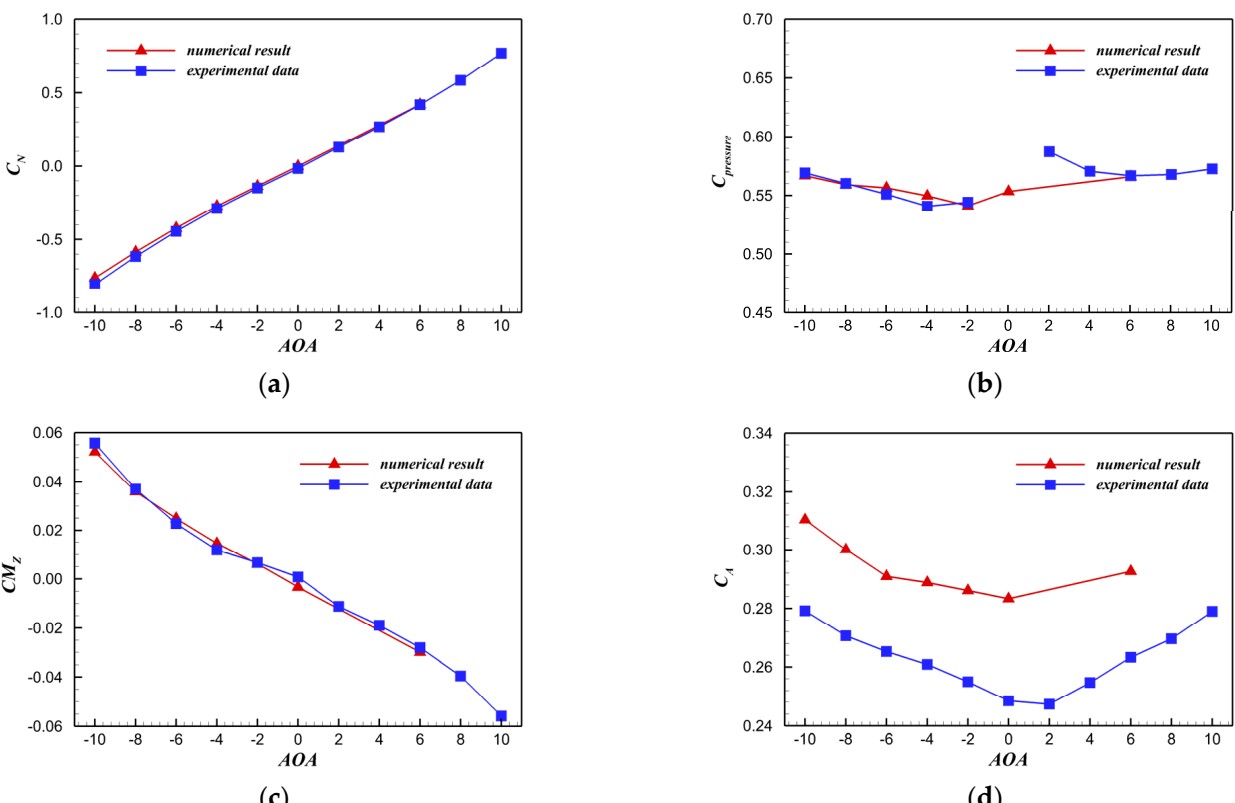

**Figure 24.** Measured and calculated aerodynamic performance at $Ma = 5$. (**a**) normal force coefficient; (**b**) center-of-pressure coefficient; (**c**) pitching moment coefficient; (**d**) axial force coefficient.

Figure 25 presents the comparison of observed aerodynamic performance with and without jet at $Ma = 5$. The figure reveals that the difference in the normal force coefficient is quite little, but the center-of-pressure coefficient of the model dramatically varies. The error of the center-of-pressure coefficient is high, because the model installation has a modest angle bow at $AOA = 0°$ and is impacted by the zero drift of the balance. As a result, it has been deleted. The jet causes the variation of pitch moment coefficient (as shown in Figure 24c) owing to the interaction of normal force and center-of-pressure coefficient, and the nosebleed air enhances the change rule of pitch moment coefficient at the small angle of attack ($-2°$ to $2°$). However, compared with without the jet, the pitch moment coefficient falls in the region of large angle of attack, such as $4°$ to $10°$ and $-4°$ to $-10°$. At $AOA = -4°$, the pitching moment coefficient without the jet is 0.01445, and that with the jet is 0.01262. The jet reduces the pitching moment coefficient by 24.5%, whereas at $AOA = -6°$, the pitching moment coefficient is reduced by 10.6%. Meanwhile, at $AOA = 6°$, the jet reduces the pitching moment coefficient by 9.2%.

To investigate the effect of altering the moment reference center on the pitch moment coefficient, the moment reference center (500 mm, 0, 0) is chosen, indicating that the reference center is closer to the pressure center. Figure 26 illustrates the observed and calculated pitching moment coefficient at $Ma = 5$. The graphic exhibits that the numerical calculation results agree with the experimental results for a wide range of angles of attack

($-8°$ to $-10°$). There are several variations when the angle of attack is small, especially at *AOA* = $-4°$. The difference might be explained by the fact that the jet reaction force has a greater impact on the pitching moment coefficient when the reference center is closer to the pressure center. However, due to the interaction of nosebleed air mass flow rate and environmental pressure at the nozzle exit, the jet generates little reaction force. Furthermore, there are deviations in balance measurements. Overall, when the reference center is (500 mm, 0, 0), the trim angle of attack of the experimental model is $7.2°$, while the numerical calculation is $5°$.

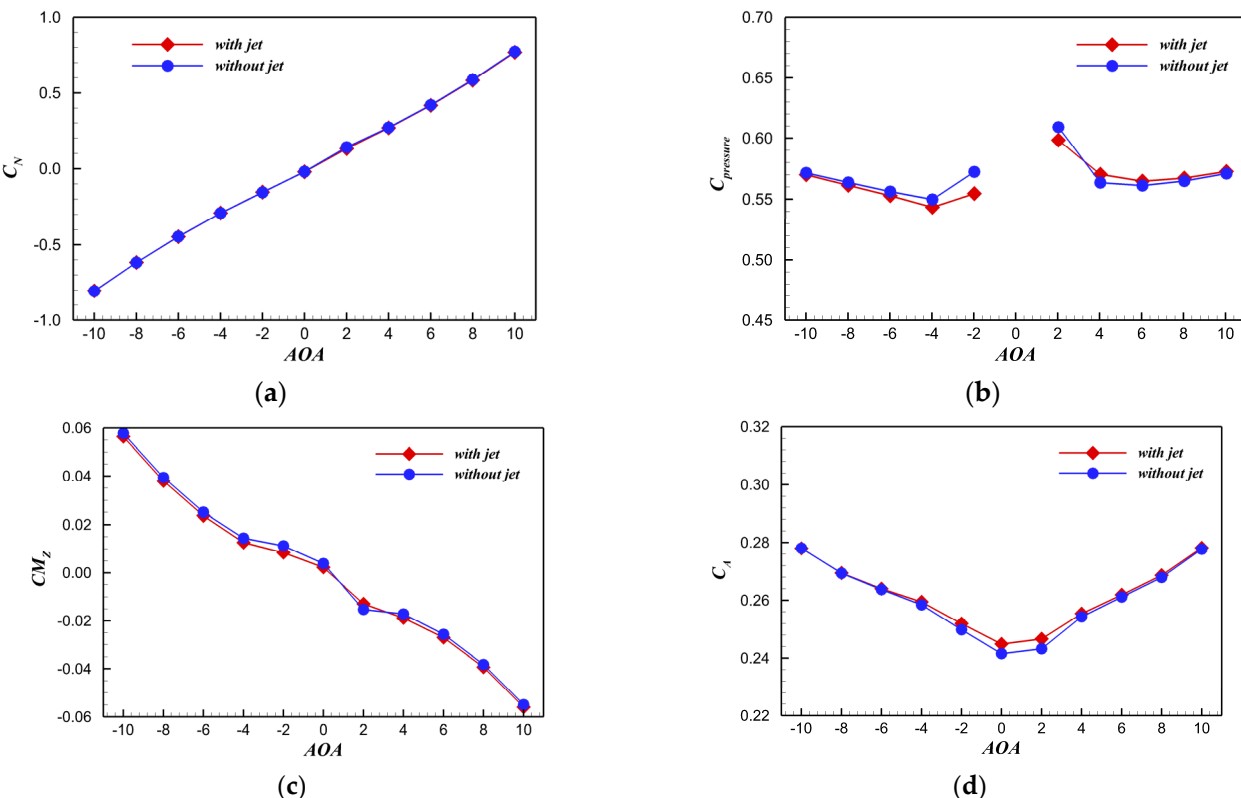

**Figure 25.** Comparison of aerodynamic performance with jet and without jet at *Ma* = 5. (**a**) normal force coefficient; (**b**) center-of-pressure coefficient; (**c**) pitching moment coefficient; (**d**) axial force coefficient.

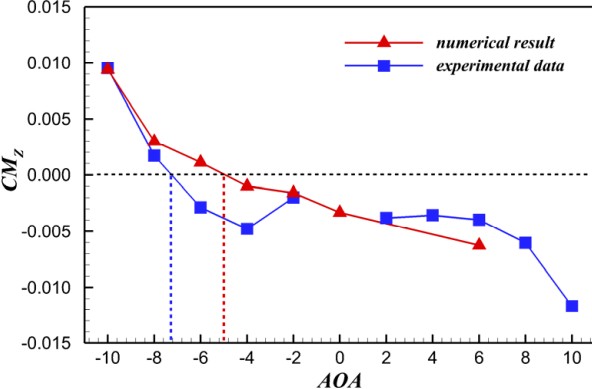

**Figure 26.** Measured and calculated pitching moment coefficient at *Ma* = 5 and moment reference center (500 mm, 0, 0).

### 5. Conclusions

A new idea of a nosebleed air jet is put forward using a hypersonic vehicle's nose stagnant high-pressure and high-temperature gas as the drive source for long-term jet control to improve maneuverability and fast reaction capabilities. This paper investigates the complicated flow characteristics of the nosebleed jet in supersonic crossflow surrounding the vehicle with a blunt head, and the jet's effect on aerodynamic coefficients is evaluated and verified by wind tunnel tests. The physical mechanism of the nosebleed jet in supersonic crossflow surrounding the vehicle for changing the flying attitude is disclosed based on the examination of the development and change of aerodynamic characteristics. The main conclusions are as follows:

(a) The new idea of employing a nosebleed air jet to control flying attitude has been confirmed through theoretical analysis, numerical simulation, and wind tunnel test. The nosebleed air jet alters the center-of-pressure coefficient, which is subsequently coupled with the interference aerodynamic force. Finally, it results in a variation in pitch moment. This gives a technological approach for hypersonic vehicles to accomplish rapid maneuverability and control.

(b) The numerical simulation findings for the normal force coefficient, center-of-pressure coefficient, and pitch moment coefficient at different angles of attack are essentially similar to the experimental results for $Ma = 5$. Because the bottom effect is not taken into account in the calculation, the axial force coefficient substantially varies. The variation of the observed and calculated axial force coefficient, on the other hand, is consistent. Meanwhile, the experiment verifies the numerical calculation method used in this study.

(c) The jet decreases the pitching moment coefficient when compared with the case without the jet. At $Ma = 5$ and $AOA = -6°$, the pitching moment coefficient is reduced by 10.6%. At $Ma = 5$ and $AOA = 6°$, the pitching moment coefficient is reduced by 9.2%.

(d) The preliminary experimental verification indicates that combining nosebleed air jet with model centroid adjustment yields an optimal trim angle of attack. The trim angle of attack of the configuration scheme in this study may be increased to 5° by adjusting the centroid.

**Author Contributions:** Conceptualization, L.Z. and K.Z.; methodology, L.Z., J.Y., T.D. and X.L.; validation, L.Z., J.Y. and J.W.; formal analysis, L.Z. and K.Z.; investigation, L.Z. and K.Z.; resources, J.Y. and J.W.; data curation, J.Y., T.D. and J.W.; writing—original draft preparation, L.Z., T.D. and J.W.; writing—review and editing, L.Z., J.Y., X.L. and K.Z.; funding acquisition, X.L. and K.Z. All authors have read and agreed to the published version of the manuscript.

**Funding:** This research was funded by a Joint Fund Project supported by the National Nature Science Foundation of China and the Civil Aviation Administration of China, grant number U2133209; Fund Project of Civil Aviation Key Laboratory of Flight Technology and Safety, grant number FZ2020ZZ01; Fund Project of Civil Aviation Flight University of China, grant numbers J2021-013 and National Nature Science Foundation of China, grant numbers 90916029.

**Institutional Review Board Statement:** Not applicable.

**Informed Consent Statement:** Not applicable.

**Data Availability Statement:** The data presented in this study are available on request from the corresponding author.

**Conflicts of Interest:** The authors declare no conflict of interest.

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
