# Peer review of "Numerical and Experimental Investigation on Nosebleed Air Jet Control for Hypersonic Vehicle"

_aerospace, doi:10.3390/aerospace10060552_

Round 1

Reviewer 1 Report

dear authors. Enclosed my comments. 

Regards. 

Author Response

Dear Reviewer,

Thank you very much for your comments and suggestions on the structure of our manuscript entitled “Numerical and Experimental Investigation on Nose Bleed Air Jet Control for Hypersonic Vehicle” (Manuscript ID: aerospace-2388496). Those comments and suggestions are all valuable and very helpful for revising and improving our manuscript, as well as the important guiding significance to our researches. We have revised the manuscript, and would like to re-submit it for your consideration. Revisions to the manuscript are marked up using the “Track Changes” function of MS Word. Point by point responses to the comments and suggestions are as flowing:

Comment 1:(The literature review concerning inlet super/hypersonic technologies addressed wit CFD approaches is minimal. In particular, authors should discuss pros and cons about CFD modeling in hypersonic condition and should provide additional references concerning design practice using CFD as a major ingredient. I suggest to have a look at the following papers: https://doi.org/10.2514/1.32187

https://doi.org/10.3390/en15082811

https://doi.org/10.24200/SCI.2016.3928

https://doi.org/10.2514/6.2021-0858)

Answer:Thank you very much for your comments and suggestions. That is very important to improve our paper writing, many pros and cons about CFD modeling, grid sensitivity test, and near wall resolutions have been revised.

Comment 2:(Author should better discuss the computational framework. In particular, no indications are provided as far the mesh characteristics, near wall resolutions…and just one mesh is adopted. I know that results well fit experiments, but the authors should convince the reader that this is not just a causality.)

Answer:Considering the comment, we have added the grid sensitivity test, near wall resolutions, etc (in “3.1. Treatment of Numerical Accuracy”) in the revised manuscript.

Comment 3:(Turbulence model is not stated.)

Answer:Considering the comment, the turbulence model statement has been added in the revised manuscript.

Comment 4:(Jet bleed is a typical unsteady phenomenon. Authors should comment why they select steady RANS and not performing unsteady calculations.)

Answer:Lateral jet is a typical unsteady phenomenon. In this manuscript, based on the lateral jet control principle, a new idea of nose bleed air jet with strong coupled internal and external flow is put forward to improve the maneuverability and fast reaction capabilities of hypersonic vehicles. At present, the main purpose is to explore the feasibility of employing a nose bleed air jet to control flight attitude. We focus on the flow field and aerodynamic performance variations of the nose bleed air jet induced by a change in flying attitude. So, the steady RANS have be solved in the manuscript. The influence of unsteady nose bleed air jet on the flow field and aerodynamic performance will be explored in the following research.

Comment 5:(Vortex shedding description in Figure 8 and 9 is not properly highlighted. I suggest to trace the vorticity contours as well zooming on the vortex shedding region.)

Answer:Thank you very much for your comments and suggestions. That is very important to improve our paper writing. At present, the main purpose is to explore the feasibility of employing a nose bleed air jet to control flight attitude. We focus on the aerodynamic performance variations of the nose bleed air jet induced by a change in flying attitude. The current vortex shedding description may explain the variation of the flow field. Development of vortex caused by nose bleed air jet will be studied in the following unsteady research.

Comment 6:(Figures 11 and 12 are very difficult to be interpreted. Please provide a better visualization of the curves.)

Answer:Considering the comment, we have adjusted the figures in the revised manuscript.

Comment 7:(Figures 24 to 27 can be combined in panels.)

Answer:We have combined Figures 24 to 27 according to the suggestion.

Comment 8:(The same for Fig 28 to 31.)

Answer:We have combined Figures 28 to 31 according to the suggestion.

We tried our best to improve the manuscript and made some changes in the manuscript. These changes will not influence the content and framework of the paper. We appreciate for your warm work earnestly, and hope that the correction will meet with approval.

If there are any problems or questions about our paper, please do not hesitate to let us know.

Once again, thank you very much for your comments and suggestions.

Sincerely,

Dr. Lin ZHANG

E-mail: harrisonzhang@163.com

Reviewer 2 Report

The authors presented a Numerical and Experimental Investigation on Nose Bleed Air Jet Control for Hypersonic Vehicle. The results are interesting, and the paper has a good scientific soundness. The paper can be accepted for publication after addressing the following comments:

The main findings are to be mentioned in the abstract.

The introduction is relatively short and may be extended.

The novelty of the paper is to be clearly stated.

The solved governing equations are to be presented.

The used turbulence model is to be justified.

Have you solved time dependent or independent equation?

The boundary conditions are to be expressed mathematically.

What are the dimensions of the computational domain.

A figure presenting the computational domain is to be added.

A figure presenting the used grid is to be added.

A grid sensitivity test is to be performed.

More details on the measurement techniques and data acquisition system are to be provided.

An experimental uncertainty study is to be performed.

Why are no flow visualisation presented?

 How are the normal force, axial force and pitching moment evaluated?

How can you explain the difference between the numerical and experimental results.

The paper is to be checked against misprints and grammatical mistakes.

Author Response

Dear Reviewer,

Thank you very much for your comments and suggestions on the structure of our manuscript entitled “Numerical and Experimental Investigation on Nose Bleed Air Jet Control for Hypersonic Vehicle” (Manuscript ID: aerospace-2388496). Those comments and suggestions are all valuable and very helpful for revising and improving our manuscript, as well as the important guiding significance to our researches. We have revised the manuscript, and would like to re-submit it for your consideration. Revisions to the manuscript are marked up using the “Track Changes” function of MS Word. Point by point responses to the comments and suggestions are as flowing:

Comment 1:(The main findings are to be mentioned in the abstract.)

Answer:We have adjusted this part according to the suggestion. In the revision, the main findings have been added in the abstract.

Comment 2:(The introduction is relatively short and may be extended.)

Answer:Considering the comment, we have extended the introduction in the revised manuscript.

Comment 3:(The novelty of the paper is to be clearly stated.)

Answer:Considering the comment, we have clearly stated the novelty in the conclusions of the revised manuscript again.

Comment 4:(The solved governing equations are to be presented.)

Answer:Considering the comment, we have added the governing equations in the revised manuscript.

Comment 5:(The used turbulence model is to be justified.)

Answer:Numerical and experimental researches on lateral jet flow have been conducted for many years. There are many achievements worth using for reference. In this paper, a new idea of nose bleed air jet is put forward to improve the maneuverability and fast reaction capabilities, which uses hypersonic vehicle’s nose stagnant high-pressure and high temperature gas as the drive source for long-term jet control. So, we choose commonly used and validated turbulence model in lateral jet researches for calculation.

Comment 6:(Have you solved time dependent or independent equation?)

Answer:According to the lateral jet control principle, a new idea of nose bleed air jet with strong coupled internal and external flow is put forward to improve the maneuverability and fast reaction capabilities of hypersonic vehicles. In the manuscript, the main purpose is to explore the feasibility of employing a nose bleed air jet to control flight attitude. We focus on the flow field and aerodynamic performance variations of the nose bleed air jet induced by a change in flying attitude. So, time independent equations have be solved.

Comment 7:(The boundary conditions are to be expressed mathematically.)

Answer:Considering the comment, we have added the boundary conditions in the revised manuscript.

Comment 8:(What are the dimensions of the computational domain.)

Answer:Three-dimensional calculation is employed in the computational domain, it has been marked in red at the first paragraph of “2.3 Computational Fluid Dynamics Method and Verification”. Considering the comment 9, we have added the computational domain and grids (as shown in Fig.6) in the revised manuscript.

Comment 9:(A figure presenting the computational domain is to be added.)

Answer:Considering the comment, we have added the computational domain and grids (as shown in Fig.6) in the revised manuscript.

Comment 10:(A figure presenting the used grid is to be added.)

Answer:Considering the comment, we have added the computational domain and grids (as shown in Fig.6) in the revised manuscript.

Comment 11:(A grid sensitivity test is to be performed.)

Answer:Considering the comment, we have added the grid sensitivity test (in “3.1. Treatment of Numerical Accuracy”) in the revised manuscript.

Comment 12:(More details on the measurement techniques and data acquisition system are to be provided.)

Answer:A detailed measurement techniques and data acquisition system has been added in the revised manuscript.

Comment 13:(An experimental uncertainty study is to be performed.)

Answer:A detailed discussion of accuracy has been added in the revised manuscript.

Comment 14:(Why are no flow visualisation presented?)

Answer:In the manuscript, a Φ300 mm color schlieren system has been used to collect schlieren photographs of with jet and without jet during the tests.

Comment 15:(How are the normal force, axial force and pitching moment evaluated?)

Answer:The six-component balance has been calibrated before the tests. If the calibration resulted in even lower values, the minimum accuracy is set to 0.05% based on previous experience. Furthermore, it is important to note that the accuracy of aerodynamic coefficients is also influenced by the accuracy of flow parameters. A detailed discussion of accuracy has been added in the revised manuscript.

We tried our best to improve the manuscript and made some changes in the manuscript. These changes will not influence the content and framework of the paper. We appreciate for your warm work earnestly, and hope that the correction will meet with approval.

If there are any problems or questions about our paper, please do not hesitate to let us know.

Once again, thank you very much for your comments and suggestions.

Sincerely,

Dr. Lin ZHANG

E-mail: harrisonzhang@163.com

Reviewer 3 Report

See attached file.

In general, the English are good and understandable.  Only some improvement on some specific terms is needed which is already mentioned in the comments attached.

Author Response

Dear Reviewer,

Thank you very much for your comments and suggestions on the structure of our manuscript entitled “Numerical and Experimental Investigation on Nose Bleed Air Jet Control for Hypersonic Vehicle” (Manuscript ID: aerospace-2388496). Those comments and suggestions are all valuable and very helpful for revising and improving our manuscript, as well as the important guiding significance to our researches. We have revised the manuscript, and would like to re-submit it for your consideration. Revisions to the manuscript are marked up using the “Track Changes” function of MS Word. Point by point responses to the comments and suggestions are as flowing:

Comment 1:(How do you get equation 1 on page 3 of the manuscript?)

Answer:According to the principle of pitot tube measurement parameters, the relationship between the flow parameters before and after the normal shock wave is deduced and analyzed by using the basic equation of one-dimensional steady flow, and Formula 1 can be obtained. In hypersonic wind tunnel tests, the total pressure after the shock wave is often measured by the pitot tube, and the total pressure before the shock wave is added to determine the incoming flow Mach number. In the supersonic flow, the total pressure behind the shock wave can also be measured by the pitot tube, and adds the static pressure before the shock wave to determine the incoming flow Mach number. In the manuscript, we know the incoming flow Mach number, the ratio of the stagnation pressure following the shock wave (P2*) to the incoming flow static pressure (p0) can be obtained according to Formula 1. The ratio means the gas source.

Comment 2:(What do you mean by throttle=6 mm at the last paragraph on page 4?)

Answer:We are very sorry for our incorrect writing. For the “throttle=6 mm”, we mean the throat diameter (hthroat). In the revision, we have modified it.

Comment 3:(On page 8 line 251, you referred to Fig. 9(1) while Figure 9 has no subfigure 1.)

Answer:We are very sorry for our incorrect writing. For the “Fig. 9(1)”, we mean Fig. 9(a). In the revision, due to the addition of figures, we have modified it to Fig. 10(a).

Comment 4:(In Figure 10, why at φ=45° the curve is separate from the other curves before reaching the jet location?)

Answer:At AOA=0°, the wall pressure distributions of different φ sections are identical before reaching the jet location. Due to the boundary layer asymmetry at AOA=6° and -6° (as shown in Fig.10(a) and Fig.10(e)), the wall pressure distribution of φ=45° section is separate from the other curves before reaching the jet location. An additional discussion has been added in the revised manuscript.

Comment 5:(In Figure 13, please identify the separation and reattachment lines.)

Answer:Considering the comment, we have added the separation and reattachment lines (as shown in Fig.13) in the revised manuscript.

Comment 6:(Can you provide your grid on the surface and around the body?)

Answer:Considering the comment, we have added the computational domain and grids (as shown in Fig.6) in the revised manuscript.

Comment 7:(What is the uncertainty in the numerical calculation? On page 12 lines 318-320, you are talking about 0.6% and 0.2% change due to the existence of the side jet. For this number to make sense, you need to provide your numerical uncertainty. Same for lines 325-326 on page 12.)

Answer:Considering the comment, we have added the grid sensitivity test, near wall resolutions, etc (in “3.1. Treatment of Numerical Accuracy”, and “Table 5 relative accuracy of six-component measurements”) in the revised manuscript.

Comment 8:(On lines 339-340 on page 13, you said “the jet causes an extremely significant difference in the pitching moment coefficient when compared to the case without jet”. From Figure 17, the changes are about 0.005. Can you explain how this change in the pitching moment can change the direction of a vehicle considering the moment of inertia of the vehicle?)

Answer:We are very sorry for our incorrect writing. In the revised manuscript, we have modified it. The jet reaction force has a greater impact on the pitching moment coefficient when the reference center is closer to the pressure center. The preliminary experimental verification indicates that combining nose bleed air jet with model centroid adjustment yields an optimal trim angle of attack. The trim angle of attack of the configuration scheme in this study may be increased to 5° by adjusting the centroid.

Comment 9:(In the last paragraph on page 13, what is the Φ on lines 346 and 349? Similarly on page 15 lines 365 and 370.)

Answerï¼šΦ is the diameter. “Φ0.5m hypersonic wind tunnel (FL-31)” means that the nozzle exit of hypersonic wind tunnel has a diameter of Φ=0.5m. “a Φ300 mm color schlieren system” means that the optical observation window of color schlieren system has a diameter of Φ=300 mm.

Comment 10:(On page 16 line 397, you mentioned that “the bow shock wave gets close to the model”. It is very hard to see that the shock gets closer to the body from Figure 22. Please quantify the change in the text.)

Answer:Considering the comment, we have quantified the change in the revised manuscript.

Comment 11:(On page 16 line 411, what do you mean by “the influence of the model bottom”? Do you mean that the numerical calculation considers only half of the body or are you talking about the base drag?)

Answer:For the “the influence of the model bottom”, we mean the base drag. In the revision, we have modified it.

Comment 12:(In Figure 25, what is the reason that at α= 2° the numerical and experimental data are getting distance? Please explain in the text.)

Answer:An explanation has been added in the revised manuscript.

Comment 13:(On page 18 line 432, please provide the data for with/without jet for α=-4° so the 24.5% make sense. It is impossible to distinguish the data in the graph.)

Answer: A data has been provided in the revised manuscript.

Comment 14:(On page 19, lines 446 and 447, you talk about the agreement between the experimental and numerical results for -4° to -10°. However, from Figure 32, there is no agreement for -2° to -8° and from 2° to 6°. For example, the difference between the data at α=-4° is about 80%. Please explain.)

Answer:We are very sorry for our negligence of the data checking. In the revised manuscript, we have modified it, and added the explanation.

Comment 15:(On page 19, lines 448 to 450, you mentioned that the trim angles are 6 and 4 degrees while on the graph in Figure 32, the trim angles are more like -7.5 and -5 degrees. Please clarify.)

Answer:We are very sorry for our negligence of the data checking. According to the graph, when the reference center is (500mm, 0, 0), the trim angle of attack of the experimental model is 7.2°, while the numerical calculation is 5°. In the revision, we have modified it, and a detailed data checking has been carried out.

Comment 16:(When you are talking about “pressure center coefficient” do you mean pressure coefficient at the center plane or the center of pressure?)

Answer:We are very sorry for our incorrect writing. For the “pressure center coefficient”, we mean center-of-pressure coefficient. In the revision, we have modified the phrase.

We tried our best to improve the manuscript and made some changes in the manuscript. These changes will not influence the content and framework of the paper. We appreciate for your warm work earnestly, and hope that the correction will meet with approval.

If there are any problems or questions about our paper, please do not hesitate to let us know.

Once again, thank you very much for your comments and suggestions.

Sincerely,

Dr. Lin ZHANG

E-mail: harrisonzhang@163.com

Round 2

Reviewer 1 Report

Dear authors, I am fine with this version of the paper
Regards

Author Response

Dear Reviewer,

Thank you very much for your comments and suggestions on the structure of our manuscript entitled “Numerical and Experimental Investigation on Nose Bleed Air Jet Control for Hypersonic Vehicle” (Manuscript ID: aerospace-2388496). Those comments and suggestions are all valuable and very helpful for revising and improving our manuscript, as well as the important guiding significance to our researches.

If there are any problems or questions about our paper, please do not hesitate to let us know.

Once again, thank you very much for your comments and suggestions.

Sincerely,

Dr. Lin ZHANG

E-mail: harrisonzhang@163.com

Reviewer 2 Report

Accept in present form

Author Response

(The authors gave the same response as above.)

Author Response

Dear Reviewer,

Thank you very much for your comments and suggestions on the structure of our manuscript entitled “Numerical and Experimental Investigation on Nose Bleed Air Jet Control for Hypersonic Vehicle” (Manuscript ID: aerospace-2388496). Those comments and suggestions are all valuable and very helpful for revising and improving our manuscript, as well as the important guiding significance to our researches. We have revised the manuscript, and would like to re-submit it for your consideration. Revisions to the manuscript are marked up using the “Track Changes” function of MS Word. Point by point responses to the comments and suggestions are as flowing:

Comment 1:(This is the continuation of Comment 7 on the original paper. Thank you for adding the grid sensitivity to the paper. However, based on the number reported in the grid sensitivity, the 0.6% and 0.2% changes in the normal force are less than the sensitivity of your calculation and thus are not reportable. The same is true for reporting 1.3% and 1.2% changes in the axial force.)

Answer:In the manuscript, the calculational results already contain the numerical uncertainty. Then, the comparisons of normal force and axial force coefficient with jet and without jet are exhibited. For the nose bleed air jet, it is different from the traditional lateral jet using attitude control motor. The diameter of the bleed channel is 12mm. The inner channel’s mass flow rate is extremely low. In this way, the jet generates very little thrust. The aerodynamic force is mainly formed by jet interference rather than the jet itself. An additional discussion has been added in the revised manuscript.

Comment 2:(Regarding my Comment 8 on the original paper, your answer still does not remove my concern. I understand your point about the effect of the reference center. However, I am still wondering how significant is the 0.005 change in the pitching moment coefficient. To be able to change the flight direction, the pitching moment should overcome the moment of inertia of the vehicle. Do you think (and please explain) how this small change can affect the trajectory of the vehicle? This explanation is required to justify your statement on line 416 on page 16. In this sentence, you are describing that “the jet causes an significant difference in the pitching moment coefficient when compared to the case without jet”.)

Answer:A data has been provided in the revised manuscript. The value of pitching moment coefficient is small. Compared with without jet, the jet reduces the pitching moment coefficient by 21.1%.

Similar to the traditional lateral jet, the bow shock surface caused by the nose bleed air jet will extend to both sides of the nozzle position to generate interference aerodynamic force, which is superimposed with the force generated by the jet to form the actual lateral force. The jet interaction may enlarge or reduce the lateral static thrust, and even lead to the thrust reversal and the jet control failure.

In this manuscript, the cross-sectional area of the bleed channel is fixed (its diameter is 12mm). The inner channel’s mass flow rate is extremely low. As a result, the jet generates very little thrust. In the following study, the difference in aerodynamic performance between with jet and without jet may be improved further by increasing the mass flow rate of the inner channel or modifying the jet nozzle.

Comment 3:(In line 205 on page 6, the transpose symbol should be T or t, not τ.)

Answer:We are very sorry for our incorrect writing. In the revised manuscript, we have modified it.

Comment 4:(There are some “an” and “a” that should be corrected in the text. For example, on line 416 on page 16, “an significant” should be “a significant”.)

Answer:We are very sorry for our incorrect writing. In the revised manuscript, we have modified it, and the uses of “an” and “a” have be corrected.

Comment 5:(Due to changes in figures numbering, there are some figures that combine into one and thus you should make sure that you don't recall one single figure as “Figs”. One example is on page 19 line 495 where “Figs. 24” should be “Fig 24”.)

Answer:We are very sorry for our negligence of those problems, and have made correction in manuscript.

Comment 6:(In line 505 of page 19, “instable” should be “unstable”.)

Answer:We are very sorry for our incorrect writing. In the revised manuscript, we have modified it.

We tried our best to improve the manuscript and made some changes in the manuscript. These changes will not influence the content and framework of the paper. We appreciate for your warm work earnestly, and hope that the correction will meet with approval.

If there are any problems or questions about our paper, please do not hesitate to let us know.

Once again, thank you very much for your comments and suggestions.

Sincerely,

Dr. Lin ZHANG

E-mail: harrisonzhang@163.com

Round 3

Reviewer 3 Report

na